# Linear Projections of Teacher Embeddings for Few-Class Distillation

## Abstract

Knowledge Distillation (KD) has emerged as a promising approach for transferring knowledge from a larger, more complex teacher model to a smaller student model. Traditionally, KD involves training the student to mimic the teacher's output probabilities, while more advanced techniques have explored guiding the student to adopt the teacher's internal representations. Despite its widespread success, the performance of KD in binary classification and few-class problems has been less satisfactory. This is because the information about the teacher model's generalization patterns scales directly with the number of classes. Moreover, several sophisticated distillation methods may not be universally applicable or effective for data types beyond Computer Vision. Consequently, effective distillation techniques remain elusive for a range of key real-world applications, such as sentiment analysis, search query understanding, and advertisement-query relevance assessment. Taking these observations into account, we introduce a novel method for distilling knowledge from the teacher model's representations, which we term Learning Embedding Linear Projections (LELP). Inspired by recent findings about the structure of final-layer representations, LELP works by identifying informative linear subspaces in the teacher's embedding space, and splitting them into pseudo-subclasses. The student model is then trained to replicate these pseudo-subclasses. Our experimental evaluations on large-scale NLP benchmarks like Amazon Reviews and Sentiment140 demonstrate that LELP is consistently competitive with, and typically superior to, existing state-of-the-art distillation algorithms for binary and few-class problems, where most KD methods suffer.

## 1 Introduction

While deep neural networks have revolutionized Natural Language Processing (Devlin et al., 2018; Brown et al., 2020), Computer Vision (Simonyan & Zisserman, 2014), and other fields, their ballooning size and data demands raise challenges. Recent research (Menghani, 2021) aims to develop efficient models that excel without needing massive datasets or expensive hardware. Knowledge Distillation (KD) (Buciluǎ et al., 2006; Hinton et al., 2015) is a powerful approach for generating lightweight models by leveraging a large teacher model to guide their training. In its basic form, the student model is trained to replicate the teacher's output probabilities for each training instance. Additionally, several subsequent studies (e.g., (Romero et al., 2014; Kim et al., 2018; Passalis & Tefas, 2018; Ahn et al., 2019; Müller et al., 2020)) have proposed advanced distillation techniques that go beyond mere output probability matching and focus on encouraging the student to learn the teacher's internal representations.

While KD can significantly improve student model performance in tasks with numerous classes, its impact is less pronounced in binary classification and problems with a smaller number of classes. As Müller et al. (2020) points out, this is because when we distill knowledge using logits or high-temperature cross-entropy, the information about the teacher model's generalization patterns scales directly with the number of classes. Furthermore, Knowledge Distillation research has primarily focused on Computer Vision, so many sophisticated distillation techniques are not always effective or even suitable for other data modalities like Natural Language. As a result, effective distillation techniques remain elusive for a range of critical real-world applications, such as sentiment analysis, search query understanding, and advertisement-query relevance assessment.

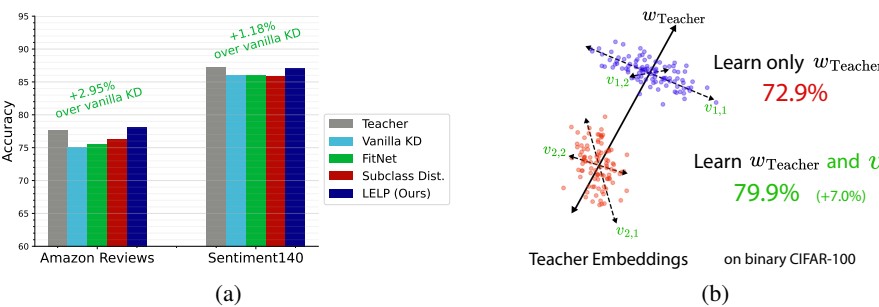

Figure 1: Learning with Embedding Projections (LELP) teaches students about subclass structure (shown by $v_{c,i}$ on (b)) via linear projections. As seen by (a), LELP outperforms existing algorithms over large scale real-world NLP datasets such as Amazon Reviews (5 classes, 500k examples) and Sentiment140 (binary, 1.6 million examples) achieving an improvement of 1.85% and 0.88%, respectively, over the best baseline. In fact, in the former case, the LELP-trained student outperforms even the teacher, **which contains over 20x the number of parameters**.

In light of these observations, in this paper we propose a novel approach for capturing information from the teacher's representations, which we call Learning Embedding Linear Projections (LELP). At a high level, it works by extracting knowledge form the teacher's last-layer representations (embeddings) and converting it into pseudo-subclasses via linear projections. The student model is then trained on these pseudo-subclasses, using a single unified cross-entropy loss.

Our approach leverages recent findings about the structure of final-layer representations (embeddings) in deep learning models and, in particular, aspects of the phenomenon known as *Neural Collapse* (Papyan et al., 2020; Fang et al., 2021; Yang et al., 2023). Similar in spirit to Subclass Distillation (Müller et al., 2020), which uses a modified and retrained teacher model to identify hidden patterns, our method achieves improved student performance, particularly in finetuning Language Models for tasks with a few number of classes. Crucially though, and unlike Subclass Distillation, our approach does not require any retraining of the teacher model. Finally, a key advantage of LELP is its flexibility in bridging diverse model architectures, making it uniquely versatile for teacher-student learning scenarios.

Our contributions can be summarized as follows:

1. Motivated by recent insights into the Neural Collapse phenomenon, we demonstrate that the invention of pseudo-subclasses through unsupervised clustering of teacher embeddings can enhance distillation performance in binary and few-class classification tasks. However, the efficacy of this approach is contingent upon the specific clustering algorithm employed.

2. Through empirical evaluation, we observe that carefully calibrated linear projections consistently achieve high performance, leading us to introduce LELP: a novel method for enhancing Knowledge Distillation. LELP is modality-independent, producing particularly strong results in NLP tasks and situations where the teacher and student architectures differ.

3. Empirical evaluations on large-scale NLP benchmarks like Amazon Reviews (5 classes, 500k examples, achieving an improvement of 1.85% over the best baseline) and Sentiment140 (binary, 1.6 million examples, showing a 0.88% improvement over the best baseline) validate that LELP is consistently competitive with, and typically superior to, existing SOTA distillation algorithms for binary and few class problems, where most KD methods suffer.

4. We show that LELP possesses several advantageous characteristics, including conveying a large amount of information per example (data efficiency), converging faster than Vanilla KD, and can provide substantial improvements in semi-supervised KD scenarios.

### 1.1 ORGANIZATION OF THE PAPER

In Section 2 we discuss related work, including the baselines we considered in this work and the Neural Collapse phenomenon. In Section 3 we present our method. In Section 4 we present our experiments. Specifically, in Section 4.2 we present experiments demonstrating that inventing pseudo-subclasses via clustering of the teacher's model's embeddings can improve distillation effectiveness,

and in Section 4.3 we compare LELP with other distillation baselines. In Section 5 we discuss the limitations of our work. In Appendix A we discuss the broader impact of our work. In Appendix B we present our experimental results corresponding to binary classification tasks with subclass structure in detail. In Appendix C we provide ablations on the design choices behind LELP. In Appendix D we give a detailed description of the NLP datasets we used. In Appendix E we present results regarding the applicability of LELP to multiclass classification tasks with a moderate number of classes. In Appendix F we study additional properties of LELP, including its relative performance in the *semi-supervised setting* (Chen et al., 2020; Stanton et al., 2021; Iliopoulos et al., 2022; Baykal et al., 2023; Kontonis et al., 2023), its data efficiency and its training speed. In Appendix G we provide pseudo-code for our method. Finally, in Appendix H we provide implementation details.

## 2 RELATED WORK

**Knowledge Distillation.** Most of the literature on Knowledge Distillation has been focused on the *fully supervised* setting, i.e., when distillation is performed on the labeled training data of the teacher model rather than on new, unlabeled data — see e.g. the original paper (Hinton et al., 2015). Specifically, when training the student one typically uses a convex combination of the standard cross-entropy loss $\mathcal{L}_{\mathrm{CE}}$ with respect to the ground-truth labels, and the *distillation loss* $\mathcal{L}_{\mathrm{distill}}$:

$$\mathcal{L}_{\mathrm{student}} = \alpha \mathcal{L}_{\mathrm{CE}} + (1 - \alpha)\mathcal{L}_{\mathrm{distill}}. \tag{1}$$

In the original paper (Hinton et al., 2015) the distillation loss encourages the student model to mimic to replicate the teacher's output probabilities, potentially after scaling the logits of both models using a temperature-scalar. (The value of the temperature is typically larger than 1, and it is used to emphasize the differences between the probabilities of wrong answers that would all be very close to zero at temperature 1.)

In addition to focusing on the teacher model's final outputs, follow-up methods, like those proposed in (Romero et al., 2014; Ahn et al., 2019), also consider the teacher model's internal representations, often in the form of its embeddings. These methods encourage the student model to mimic not only the teacher's final predictions but also its internal representations. The simplest, yet often highly effective, example of this approach is *Embedding Distillation* (Romero et al., 2014). In this method, an additional term, called the "embedding-loss term", is added to the distillation loss. This term measures the difference between the embeddings produced by the teacher and student models. The weight of this term can be adjusted using a hyperparameter. Specifically, one adds the term

$$\mathcal{L}_{\mathrm{Embedd}} = \frac{1}{n}\sum_{i=1}^{n} \|f^T(x_i) - W f^S(x_i)\|_2, \tag{2}$$

where $x_1, \ldots, x_n$ denotes the current batch of examples, $f^T(x_i), f^S(x_i)$ the embeddings of the teacher and student model corresponding to example $x_i$, respectively, and $W$ is a learnable projection matrix to match the dimension of the teacher/student learned during the distillation phase. Romero et al. (2014) proposes ways of modeling and pretraining $W$ (and also potentially distilling from other teacher-layers), while Ahn et al. (2019) proposes losses that minimize the mutual information between the teacher and student embeddings (instead of considering the $\ell_2$ loss). Finally, Relational Knowledge Distillation (Park et al., 2019) transfers mutual relations of data examples instead, e.g., they authors introduce a loss that penalizes structural differences in "anglewise" relations. Notably, Relational KD can naturally handle mismatches in teacher-student embedding dimensionality without introducing learnable projections.

The Subclass Distillation method (Müller et al., 2020) presents an approach that is closely related to ours. Here, the teacher is forced to divide each class into many pseudo-subclasses that it invents via appropriate training, and then the student is trained to match the subclass probabilities. The method is designed for few-class classification, excelling in such scenarios. Our method shares Subclass Distillation's use of invented subclasses for student knowledge transfer, but eliminates the need for teacher retraining and extensive hyperparameter tuning. (Note that the process of optimizing hyperparameters for Subclass Distillation necessitates retraining the teacher model repeatedly with varying loss function configurations, as outlined in equations (7) and (8) of the reference paper by Müller et al. (2020). However, this iterative retraining becomes *excessively computationally intensive and impractical* when dealing with large teacher models.) As we show in Section 4, LELP achieves performance that is always on par with, and typically exceeding, Subclass Distillation.

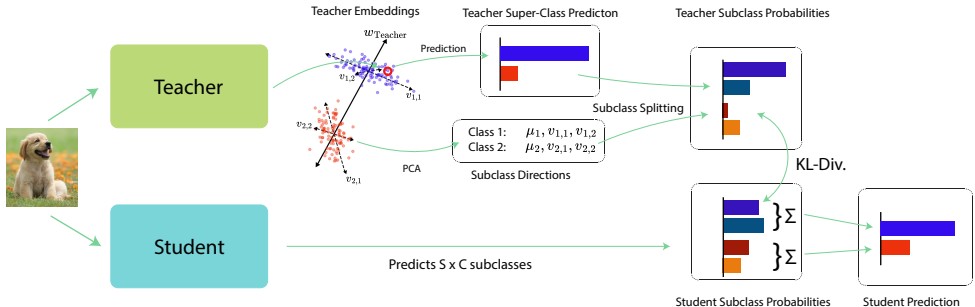

Figure 2: Schematic of the Learning with Embedding Projections (LELP) algorithm. LELP decomposes Teacher predictions into subclasses via a PCA decomposition, and trains a student on these subclasses. For predictions, subclasses are summed together back into their original classes.

In our study we show that LELP is consistently competitive with the above methods, and that it is typically superior to them when it comes to classification problems with a few number of classes (especially in the NLP domain). We also explore two more baseline methods that have found success in Vision tasks: Contrastive Representation Distillation (CRD) Tian et al. (2019) and Decoupled Knowledge Distillation (DKD) Zhao et al. (2022). While CRD excels in Vision multi-class tasks, its reliance on data augmentation for contrastive learning limits its applicability to other domains. Indeed, in Vision classification tasks the standard "contrastive-learning" approach would be to generate variations of every image (e.g, via rotations), and then consider as positive any pair of examples (original or modified) that come from the same source. However, data augmentation of, say Natural Language examples, is not straightforward and this impairs the performance of the method. (See Appendix H for more details.) DKD has also demonstrated efficacy in multi-class Vision tasks, but its performance is notably compromised in scenarios with fewer classes, rendering it inappropriate for the specific focus of our investigation. In particular, in the context of binary classification, DKD is mathematically equivalent to the standard Vanilla KD approach. We demonstrate that LELP significantly outperforms both CRD and DKD in text classification tasks.

**Neural Collapse.** Papyan et al. (2020) identified a phenomenon called *Neural Collapse* that occurs in the final training phase (after achieving zero training error) of deep neural networks. This phenomenon refers to specific structural properties observed in the last layer's representations. The simplest of these properties is known as *variability collapse*. In a nutshell, as a network is trained extensively, it tends to group together all training samples with the same label (almost) into a single point in its final layer.

Our approach is inspired by a recent paper Yang et al. (2023) which provides evidence that further refines the above description. In particular, it shows that while the final layer representations appear collapsed, they retain crucial fine-grained structure. This structure, though subtle, accurately reflects the inherent characteristics of the input distribution. As an illustrative example, the authors consider a model trained to classify images in the CIFAR-10 Krizhevsky et al. (2009) dataset using only 5 coarse-graned labels (by combining two classes into one superclass). Remarkably, even after this training, they can still recover the original, more detailed 10-class system by applying unsupervised clustering techniques to the model's internal representations. In light of this intriguing observation, and motivated by the findings of Müller et al. (2020) that creating meaningful pseudo-subclasses aids student knowledge transfer, we present an unsupervised technique to extract knowledge from teacher embeddings using linear projections to form pseudo-subclasses. Our approach, which we compare to other clustering methods (including the one in Yang et al. (2023)) in Table 3 in Section 4.2, consistently shows superior performance.

## 3 LEARNING FROM EMBEDDING LINEAR PROJECTIONS (LELP)

Since large models develop embeddings that hold information not captured in their output probabilities (Yang et al., 2023), we aim to transfer this knowledge to student networks while adhering to three desiderata:

**Modality-independent**. We aim to develop a data-agnostic method, meaning its performance remains consistent regardless of the underlying data type. The method should not be reliant on characteristics specific to certain data modalities (for instance, the ability to easily perform data augmentation).

**Compatibility between differing Student/Teacher architectures**. Embedding Distillation, the most straightforward method for learning the teacher's embedding information, only works out-of-the-box when the student and teacher have matching embedding dimensions. Otherwise, a learnable projection layer is required to match the student/teacher embedding dimensions, which can often harm performance (and even increase latency in certain cases). Therefore, we aim for a method that is embedding dimension agnostic.

**No Retraining the Teacher Model**. When the teacher model is significantly larger than the student model, retraining it can be prohibitively expensive. Methods which require a modified teacher training pipeline are significantly more costly to tune hyperparameters for, as not only do the teacher models have to be retrained multiple times, but in addition the student hyperparameters need to be checked against varying teacher hyparameters. This precludes methods such as Subclass Distillation (Müller et al., 2020), which contains several teacher training hyperparameters, and can even hurt the teacher's performance.

With these requirements in mind, we present Learning Embedding Linear Projections (LELP), which extracts the information in the teacher's embedding layer into pseudo-subclasses, on which the student is trained in a single unified cross-entropy loss. To motivate our method, we refer to the toy model of the learned teacher embeddings shown in Figure 2. Here, data points cluster around their respective class averages, but each cluster exhibits internal structure that holds semantically meaningful information. This structure differentiates individual items within the same class and has proven valuable for identifying subclasses when they exist Müller et al. (2020). Additionally, (Yang et al., 2023) shows that these subclasses can be separated using a linear probe (following an appropriate clustering of the embeddings space), which motivates the use of linear projections as a means of extracting further information for distillation.

LELP extracts this embedding information into a single classification loss in three steps. Firstly, we identify meaningful linear subspaces in the teacher embedding space (section 3.1). Secondly, we project the teacher embeddings into these subspaces, and use them to form pseudo-subclasses, expanding the number of classes from $C$ to $S \times C$, where $S$ is the number of linear projections we employ per class (section 3.2). Finally, we perform standard knowledge distillation, where additionally the student network's final layer outputs probabilities for $S \times C$ classes (section 3.3).

## 3.1 IDENTIFYING INFORMATIVE LINEAR SUBSPACES

The first step in LELP requires identifying linear subspaces in the teacher embedding space which contain useful information for each class cluster. Specifically, let $\{x_i^c\}_{i=1}^{N_c}$, be the $N_c$ training points in the dataset belonging to class $c$, and let $\{h_i^c\}_{i=1}^{N_c}$ be the corresponding teacher embeddings in $\mathbb{R}^D$, that is, $h_i^c = h^{\text{Teacher}}(x_i^c)$, where $h^{\text{Teacher}}(x)$ is the teacher feature extractor. We want to find the $S$ most informative linear directions in $\{h_i^c\}_{i=1}^{N_c}$. In absence of further knowledge of the structure of $\{h_i^c\}_{i=1}^{N_c}$, we opt to take the $S$ top PCA directions. This PCA decomposition yields for each class a class mean $\mu_c$ and $S$ top PCA directions: $\{v_{c,1}, v_{c,2}, \ldots, v_{c,S}\}$. Two remarks are in order.

First, observe that such a PCA can contain "redundant" information which is already captured in the teacher's output weights. Specifically, if the teacher's output weights are $\{w_i, \ldots, w_c\}$, standard knowledge-distillation will already contain all the information along these directions, meaning that further information in these directions from $v_{c,i}$ is unnecessary. Therefore, we have found that it often helps to first project $\{h_i^c\}_{i=1}^{N_c}$ onto the null-space of the teacher weights $\{w_i, \ldots, w_c\}$ before performing the PCA. Of course, in principle, the null-space could be trivial, in which case we do not apply this step. Typically though, we are working in the regime where $D \geq S + C$, so this step does apply. In particular, we have not run into the problem in the experiments of this study.

A second consideration is the imbalance of variance between PCA directions: projections onto the $v_{c,1}$ directions to contain the most variances and each subsequent will contain less. If there is significant imbalance, we find that this can lead to poor performance so we have found it is useful to apply a random rotation on the PCA directions so the variance in each direction is equal. Specifically, we use

$\tilde{V}_c = QV_c$, where $V_c = [v_{c,1}, \ldots, v_{c,S}] \in \mathbf{R}^{S \times D}$ is our PCA vectors concatenated and $Q \in \mathbb{R}^{S \times S}$ is a random orthonormal matrix. $\tilde{V}_C = [\tilde{v}_{c,1}, \ldots, \tilde{v}_{c,S}]$ contains our random rotated PCA directions, which span the same space as $V$, but each direction has the same variance in expectation.

In Appendix C we perform ablations where we compare applying LELP to simply applying PCA or Random Projections, in order to show that the above observations can indeed be beneficial in terms of the student's performance. Algorithm pseudo-code for the step of identifying the linear subspaces is provided in Algorithm 1 in appendix G.

The cost of performing the PCA is $O(N_c D^2 + D^3)$ in time and $O(D^2)$ in memory, where $D$ is the embedding dimension. In practice, the $O(N_c D^2)$ associated with forward-passing the dataset through the teacher network is the most costly compared to the $O(D^3)$ PCA cost for practical embedding dimensions, making the practical cost of the computation $O(N)$ where $N$ is the training dataset size.

## 3.2 Splitting into pseudo-subclasses

Given the set of $C$ class embedding means and $S$ PCA vectors per class, we now describe how we split the $C$ classes into $SC$ subclasses. Let $p_c^{\text{Teacher}}$ be the teacher class probability of class $c$, with temperature parameter $\tau$. That is,

$$p_c^{\text{Teacher}} = \frac{e^{z_c/\tau}}{\sum_{i=1}^{C} e^{z_i/\tau}},$$

where $z_c$ are teacher logits, i.e. $z_c = w_c^{\text{Teacher}\intercal} h - b_c$, with $w_c^{\text{Teacher}}$ the teacher weight for class $c$ and $b_c$ the bias. We split this into $S$ subclass probabilities $p_{c,1}, \ldots p_{c,S}$ where

$$p_{c,s}^{\text{Teacher}} = p_c^{\text{Teacher}} * \frac{e^{z_{c,s}/\beta}}{\sum_{j=1}^{S} e^{z_{c,j}/\beta}}.$$

where $z_{c,s} = \tilde{v}_{c,s}^{\intercal}(h - \mu_c)$. That is, we perform a tempered softmax over *subclass logits* $z_{c,s}$, where $z_{c,s}$ are given by the PCA decomposition coordinates for that class. $\beta$ is our subclass tempering parameter which is a hyperparameter in our method. We refer to this subclass splitting algorithm as `subsplit`, which takes as input the teacher embedding $h$, the PCA direction and mean vectors $\tilde{V}$ and $M$, the teacher final layer weights $W$, and temperature $\beta$. Pseudocode is in Appendix G.

## 3.3 Knowledge Distillation with subclasses

Finally, we perform standard knowledge distillation with our new $SC$ probabilities $p_{c,s}$. This requires a straightforward modification to the student architecture, in which it outputs $SC$ classes as opposed to the standard $C$ classes. We applying the standard tempered Knowledge Distillation loss as prior work, using the same temperature $\tau$ used to generate $p_{c,s}$:

$$\mathcal{L}_{\text{LELP}} = \tau^2 \mathcal{D}(p_{c,s}^{\text{Teacher}} || p_{c,s}^{\text{Student}}),$$

where $\mathcal{D}$ is a standard classification loss like Cross-Entropy or KL-Divergence. (In all our experiments in this paper we use the latter.)

With $p_{c,i}^{\text{Student}}$ using the same temperature parameter $\tau$:

$$p_{c,s}^{\text{Student}} = \frac{e^{z_{c,s}^{\text{Student}}/\tau}}{\sum_{i=1}^{C} \sum_{j=1}^{S} e^{z_{i,j}^{\text{Student}}/\tau}}.$$

At test time, we simply take class probabilities to be the sum over subclass probabilities: $p_c^{\text{Student}} = \sum_{j=1}^{S} p_{c,j}^{\text{Student}}$ and take the prediction to be the class with largest probability. By putting embedding information directly into a unified classification loss, our algorithm avoids careful balancing of training objectives such as with Embedding Distillation, in which one must carefully tune the embedding loss coefficient. Furthermore, our method is minimally invasive, requiring only minor changes to the student network and the training pipeline relative to other effective Knowledge Distillation techniques. In particular, our method avoids the need for pretraining steps of learnable projections, unlike FitNet and VID, and it also doesn't require a memory buffer (sometimes used for CRD), or retraining the teacher model like Subclass Distillation. All put together, our algorithm is given in Algorithm 3.

## 4    EXPERIMENTAL EVALUATION

In this section we present our experimental results. In Section 4.1 we describe our experimental setup. Section 4.2 presents experiments showcasing that generating pseudo-subclasses via unsupervised clustering of the teacher model's embeddings can improve distillation effectiveness. In Section 4.3 we compare LELP with other distillation baselines.

### 4.1    THE SETUP

We focus our experiments on a variety of classification tasks, using a standard distillation setup as in (Hinton et al., 2015). In order to focus solely on the effect of the distillation loss of each method, we always set $\alpha = 0$ in  equation 1. This reduces the variance between methods which may have different optimal values of $\alpha$, and reduces the hyperparameter search space.  Furthermore, in the important case of the semi-supervised setting one does not have access to ground-truth labels.

**The Architectures**    Given that the specific combination of student and teacher architectures is known to influence the effectiveness of knowledge distillation, we have chosen to evaluate various student-teacher pairings to ensure the robustness of our method.

For Vision datasets, the "ideal" distillation scenario is given by distilling ResNet-92 to ResNet-56, where both architectures and embeddings dimensions match ($D = 256$). We also consider the case of a smaller and a larger dimension with ResNet-92 ($D = 256$) and MobileNet with width and depth multiplier equal to 2 ($D = 2048$), respectively, distilling to MobileNet ($D = 1024$). The latter cases address scenarios where the embedding dimensions differ, with one scenario involving the same architecture family and the other involving different architectures. For smaller NLP datasets (LMRD, GLUE/cola, and GLUE/sst2), we consider distillation from an ALBERT-Large model ($D = 1024$) to an ALBERT-Base model ($D = 768$). For the larger scale NLP datasets (based on Amazon US reviews and Sentiment 140) we consider distillation from an ALBERT-XXL model ($D = 4096$) and an ALBERT-XL model ($D = 2048$) to (i) an ALBERT-Base model ($D = 768$); (ii) and a two-layer-MLP of width ($D = 4096$) that operates over representations generated by a (frozen) sentence-T5 encoder model of 11 billion parameters (Ni et al., 2021). The latter case addresses the scenario involving different teacher-student architectures but with the same embedding dimension. (Using a pre-trained, frozen large-scale encoder model to generate representations is a common-in-practice approach when one needs multiple lightweight models for different classification tasks.)

**The Baselines**    To establish a baseline performance, we chose well-known distillation approaches. In particular, we consider Standard Training of the model with the ground-truth labels and the cross-entropy loss, Vanilla Distillation with temperature-scaling as described in the original paper of Hinton et al. (2015), the Embedding Distillation method as described in Section 2, the general FitNet Romero et al. (2014) approach[1] , the Variational Information Distillation for Knowledge Transfer (VID) framework Ahn et al. (2019), the Relational KD approach Park et al. (2019), Contrastive Representation Distillation CRD Tian et al. (2019), Decoupled Knowledge Distillation Zhao et al. (2022) and Subclass Distillation Müller et al. (2020). As previously noted, DKD is functionally identical to Vanilla KD in the context of binary classification tasks. Consequently, for such tasks, *reported values for DKD and Vanilla KD are congruent*. For all baselines we perform a grid search over their relevant parameters and report the performance (test-accuracy and standard deviation over three trials) of the best configuration, with hyperparameters given in Appendix H.3. The best forming algorithm is shown in bold, and the second best underlined. Additionally, for every scenario in our tables, we display the average improvement of LELP compared to: (i) the top-performing baseline; (ii) the best baseline excluding Subclass Distillation; (iii) Vanilla Distillation. We report the second comparison for a couple of reasons. Firstly, optimizing Subclass Distillation's hyperparameters is significantly more costly compared to the other baselines, and it can be challenging in real-world situations (due to the requirement of retraining and storing multiple teacher models). Second, the accuracy of the teacher model in Subclass Distillation usually differs from the one used for LELP (and the other baselines). Therefore, comparing them directly might not be entirely fair.

---

[1]Embedding Distillation as described in Section 2 can be thought of as an instance of the FitNet framework.

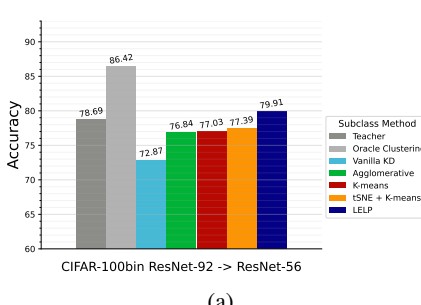 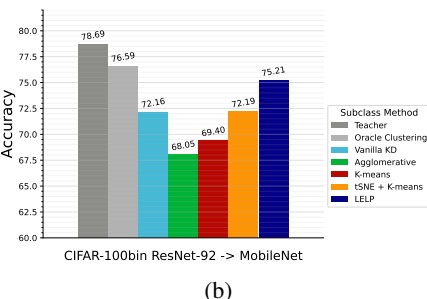

(a)                                    (b)

Figure 3: The effectiveness of different clustering techniques for creating pseudo-subclasses during knowledge distillation from a ResNet-92 teacher to (a) a ResNet-56 and (b) a MobileNet student on the binarized CIFAR-100 dataset is presented. "Oracle Clustering", where subclass structure is known a priori, serves as an upper bound and notably surpasses all other methods, even exceeding the teacher's performance in the ResNet-56 case. Among practical methods (i.e., those discovering subclass structure), LELP exhibits superior performance. Agglomerative and K-means clustering do not consistently outperform vanilla knowledge distillation, demonstrating the dependence of pseudo-subclasses effectiveness on the chosen clustering algorithm.

## 4.2 INVESTIGATING METHODS FOR INVENTING PSEUDO-SUBCLASSES

In this section we investigate the extent to which inventing pseudo-subclasses via unsupervised clustering of the teacher model's embeddings can be beneficial for distillation. Our investigation focuses on tasks with inherent subclass structure, driven by the hypothesis that class embeddings contain rich and informative structural details. Specifically, we utilize binarized versions of CIFAR-10 and CIFAR-100, assigning binary labels based on the original class labels ($y_{binary} = y_{original}\%2$). These datasets present a unique challenge for knowledge distillation, as vanilla distillation is known to underperform due to limited class label information.

Crucially, the availability of original labels ($y_{original}$) in these subclass datasets allows us to explore the "Oracle Clustering" approach. This entails training the student model on the full 10/100-way classification task for CIFAR-10/100, respectively, and subsequently treating the problem as binary classification during testing. Interestingly, the student models trained utilizing the Oracle Clustering approach exhibit the highest performance among all evaluated clustering methods. This approach not only outperforms other clustering strategies but also surpasses the performance of the original teacher network in certain cases. A t-SNE visualization of the feature embeddings learned by the student models, presented in Figure 4, offers a deeper understanding of the underlying factors contributing to this observed performance advantage. While Oracle Clustering represents an idealized scenario where subclass structure is known a priori and thus impractical for real-world datasets, its consideration underscores the potential power of inventing pseudo-subclasses as a methodological approach.

We consider several natural ways of inventing pseudo-subclasses by clustering the teacher's embedding space and we compare them with LELP. In particular, we consider three approaches: (i) Agglomerative clustering (complete linkage); (ii) K-means clustering; and (iii) K-means clustering after we first have projected the teacher's embedding space in a two-dimensional space using t-SNE (Hinton & Roweis, 2002) as proposed in (Yang et al., 2023). For these methods, we tested both hard-label (one-hot vector) and soft-label (weighted by teacher probabilities) representations of the subclasses during student training. We observed that soft-labels, even with temperature tuning, did not significantly impact performance. Therefore, we present results using only hard-label (one-hot vector) subclass representations, which can be found in Table 1 (see also Figure 3). Details on the number of clusters chosen can be found in Appendix H.3.

Analysis of Table 1 reveals several key findings: (i) LELP consistently outperforms all other clustering methods as well as Vanilla KD across all experimental scenarios; (ii) t-SNE & K-means generally demonstrate superior performance to Vanilla KD; (iii) Agglomerative clustering and K-means clustering exhibit more varied results in comparison to Vanilla KD. Thus, our findings suggest that the generation of pseudo-subclasses via clustering methodologies holds significant potential, however, the efficacy of this approach is contingent upon the specific clustering algorithm employed.

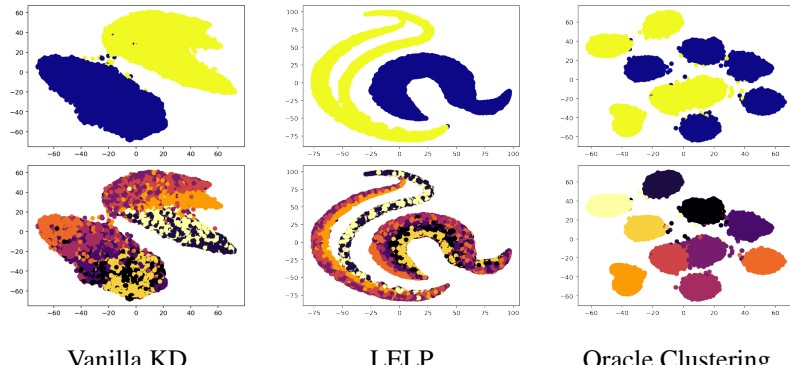

Vanilla KD          LELP          Oracle Clustering

Figure 4: The t-SNE visualization of the feature embeddings learned by student models during knowledge distillation from a ResNet-92 teacher to a ResNet-56 student on the binarized CIFAR-10 dataset. The first row depicts embeddings colored according to the two primary classes of the task, while the second row uses color to represent the underlying subclasses. Notably, the student model trained with the proposed LELP method exhibits a more intricate embedding structure compared to the student trained with standard Vanilla KD. Furthermore, the embeddings of the student trained via the "ideal" Oracle Clustering approach clearly delineate the underlying subclass structure, providing insight into its ability to surpass even the teacher's performance.

Table 1: Comparison of different unsupervised clustering approaches for inventing pseudo-subclasses.

| Teacher Architecture | ResNet92 | ResNet92 | MobileNetWD2 | ResNet92 | ResNet92 | MobileNetWD2 |
|---|---|---|---|---|---|---|
| Student Architecture | ResNet56 | MobileNet | MobileNet | ResNet56 | MobileNet | MobileNet |
| Dataset | CIFAR-10-bin | CIFAR-10-bin | CIFAR-10-bin | CIFAR-100-bin | CIFAR-100-bin | CIFAR-100-bin |
| Teacher | 96.70 | 96.70 | 93.90 | 78.69 | 78.69 | 72.63 |
| Oracle Clustering | $97.17 \pm 0.02$ | $93.23 \pm 0.28$ | $93.48 \pm 0.05$ | $86.42 \pm 0.11$ | $76.59 \pm 0.60$ | $77.03 \pm 0.1$ |
| Vanilla Distillation | $95.75 \pm 0.21$ | $92.91 \pm 0.17$ | $\underline{92.79 \pm 0.3}$ | $72.87 \pm 0.27$ | $72.16 \pm 0.39$ | $\underline{71.45 \pm 0.59}$ |
| Agglomerative | $95.48 \pm 0.08$ | $92.16 \pm 0.15$ | $92.41 \pm 0.28$ | $76.84 \pm 0.38$ | $68.05 \pm 1.80$ | $66.97 \pm 0.93$ |
| K-means | $95.76 \pm 0.09$ | $92.53 \pm 0.32$ | $92.05 \pm 0.14$ | $77.03 \pm 0.21$ | $69.4 \pm 0.87$ | $68.97 \pm 0.79$ |
| t-SNE & K-means Yang et al. (2023) | $\underline{95.92 \pm 0.19}$ | $\underline{92.97 \pm 0.07}$ | $91.20 \pm 0.52$ | $\underline{77.39 \pm 0.29}$ | $\underline{72.19 \pm 0.60}$ | $68.15 \pm 0.09$ |
| LELP (Ours) | $\mathbf{96.71 \pm 0.04}$ | $\mathbf{93.99 \pm 0.17}$ | $\mathbf{93.03 \pm 0.10}$ | $\mathbf{79.91 \pm 0.15}$ | $\mathbf{75.21 \pm 0.12}$ | $\mathbf{72.38 \pm 0.24}$ |

### 4.3 COMPARISON WITH PREVIOUS APPROACHES

In this section we compare LELP with other distillation baselines.

#### 4.3.1 WARMUP: BINARY CLASSIFICATION WITH SUBCLASS STRUCTURE

As a warmup, we first focus on on binary datasets that have an inherent subclass structure following the setting of Section 4.2. The results can be found in Table 3 in Appendix B. Among the baselines, Subclass Distillation often performs the best, consistent with the finding that subclass-splitting is effective with little label information (Müller et al., 2020). Our method is able to capture the learned structure from the teacher subclasses without retraining the more expensive teacher model and outperforms all baselines. It's worth noting that our method achieves the highest average improvement over the leading baseline when applied to the CIFAR-100bin dataset, specifically when transferring knowledge from a ResNet teacher model to a MobileNet student model. This scenario is particularly challenging due to the presence of numerous subclasses and significant differences between the teacher and student models in terms of both embedding dimensions and architectures. Interestingly, our method even outperforms Oracle Clustering in the CIFAR-10bin ResNet to MobileNet case, suggesting that it leverages structure that is learned by the teacher that is not present in the original CIFAR-10 labels.

#### 4.3.2 FEW-CLASS CLASSIFICATION WITHOUT SUBCLASS STRUCTURE

In this section we focus on datasets without subclass structure. We consider classification on six language classification tasks: the Large Movie Review dataset (Maas et al., 2011), two GLUE datasets: cola and sst2 (Wang et al., 2018), two datasets sampled from the Amazon US reviews

datasets (Datasets, 2020), and a dataset sampled from the Sentiment140 dataset (Go et al., 2009). Details are given in Appendix D. Our results are shown in Table 2. Notably, Subclass Distillation is typically the best performing baseline. LELP exceeds or performs as well as Subclass Distillation, but importantly does not require retraining the teacher, which in the age of ever growing large language models, becomes increasingly important. It is also worth noting that for the case of Amazon US Reviews-based datasets (among the largest ones considered in our study) where we distill to an ALBERT-Base model, LELP significantly outperforms even the teacher model, which contains over 20x the number of parameters.

## 5 LIMITATIONS

While our method is simple and efficient to implement, there may be limitations in using a simple linear projection on the teacher final layer embeddings to extract subclass data. Firstly, there is no reason to assume that the subclasses should be linearly separable in the teacher embedding, and it is likely that more sophisticated unsupervised clustering methods could extract richer information. Second, LELP performs when there is limited teacher logit information (such as in binary classification tasks), however larger image datasets with many classes contain sufficient information in their teacher logits, obviating the need of subclass splitting methods such as LELP. Indeed, as the number of classes increases, we anticipate LELP's performance to converge with vanilla knowledge distillation. (Consequently, we did not present experiments on datasets with a large number of classes, such as ImageNet-1k, since LELP is not designed for such scenarios.)

## 6 CONCLUSION

In this study, we have presented evidence that the creation of pseudo-subclasses via unsupervised clustering of teacher embeddings can improve distillation performance in binary and few-class classification tasks, without necessitating the retraining of the teacher model. Through empirical evaluation, we observed that linear projections consistently yield high performance, prompting us to introduce LELP. The superior performance of the "Oracle Clustering" method, where subclass structure is known a priori, suggests that the generation of pseudo-subclasses through clustering techniques has substantial promise. Consequently, future research can explore more sophisticated methods for extracting teacher embedding information, drawing insights from the expanding body of work on Neural Collapse, or investigate strategies for distilling intermediate layer embeddings using a similar approach.

## 7 REPRODUCIBILITY STATEMENT

Appendix G provides a detailed description of LELP, including pseudo-code. Implementation details for all distillation methods employed in this study, including hyperparameter choices, can be found in Appendix H. Appendix D offers a comprehensive overview of the NLP datasets used in this research. Finally, a Jupyter Notebook containing the LELP implementation and code to reproduce the binary classification results with subclass structure is available in the supplementary material.

Table 2: Experiments on Classification Tasks **without** Subclass Structure.

| Teacher Architecture | ALBERT-Large | ALBERT-Large | ALBERT-Large | ALBERT-XXL | ALBERT-XXL | ALBERT-XXL | ALBERT-XXL | ALBERT-XL |
|---|---|---|---|---|---|---|---|---|
| Student Architecture | ALBERT-Base | ALBERT-Base | ALBERT-Base | ALBERT-Base | MLP/T5(11B) | ALBERT-Base | MLP/T5(11B) | ALBERT-Base |
| Dataset | LMRD | GLUE/cola | GLUE/sst2 | Am. Reviews Bin | Am. Reviews Bin | Am. Reviews | Am. Reviews | Sentiment140 Bin |
| Teacher | 90.19 | 81.87 | 94.09 | 87.82 | 87.82 | 77.58 | 77.58 | 87.29 |
| Subclass Distillation Teacher | 90.05 | 81.49 | 93.11 | 87.71 | 87.53 | 78.45 | 78.02 | 87.45 |
| Standard Training | $87.68 \pm 0.46$ | $79.13 \pm 0.79$ | $91.32 \pm 0.12$ | $85.95 \pm 0.09$ | $87.83 \pm 1.48$ | $74.51 \pm 0.32$ | $66.44 \pm 0.23$ | $85.82 \pm 0.15$ |
| Vanilla Distillation | $88.70 \pm 0.17$ | $80.50 \pm 0.05$ | $92.39 \pm 0.24$ | $86.76 \pm 0.23$ | $88.60 \pm 0.1$ | $75.13 \pm 0.12$ | $72.30 \pm 1.19$ | $85.95 \pm 0.17$ |
| Embedding Distillation | $88.98 \pm 0.33$ | $80.66 \pm 0.65$ | $92.62 \pm 0.16$ | $86.43 \pm 0.14$ | $89.84 \pm 0.78$ | $75.28 \pm 0.07$ | $72.64 \pm 0.95$ | $85.78 \pm 0.03$ |
| FitNet | $88.75 \pm 0.15$ | $80.82 \pm 0.32$ | $92.43 \pm 0.18$ | $86.53 \pm 0.2$ | $91.40 \pm 0.5$ | $75.54 \pm 0.05$ | $75.78 \pm 0.21$ | $86.07 \pm 0.02$ |
| VID | $88.26 \pm 0.20$ | $79.83 \pm 0.62$ | $91.47 \pm 0.33$ | $83.74 \pm 0.7$ | $89.23 \pm 0.91$ | $67.48 \pm 0.81$ | $63.98 \pm 1.79$ | $85.83 \pm 0.01$ |
| Relational KD | $88.90 \pm 0.24$ | $81.24 \pm 0.43$ | $92.73 \pm 0.15$ | $86.36 \pm 0.09$ | $89.52 \pm 0.95$ | $75.12 \pm 0.39$ | $74.72 \pm 0.91$ | $86.12 \pm 0.07$ |
| DKD | $88.70 \pm 0.17$ | $80.50 \pm 0.05$ | $92.39 \pm 0.24$ | $86.76 \pm 0.23$ | $88.6 \pm 0.1$ | $72.41 \pm 0.91$ | $72.40 \pm 0.82$ | $85.95 \pm 0.17$ |
| CRD | $89.19 \pm 0.03$ | $80.79 \pm 0.21$ | $92.27 \pm 0.48$ | $85.79 \pm 0.11$ | $88.12 \pm 0.91$ | $74.7 \pm 1.32$ | $61.08 \pm 1.8$ | $84.60 \pm 0.69$ |
| Subclass Distillation | $\mathbf{89.24 \pm 0.31}$ | $80.85 \pm 0.1$ | $\mathbf{92.85 \pm 0.15}$ | $87.34 \pm 0.1$ | $90.38 \pm 0.82$ | $76.23 \pm 0.50$ | $77.27 \pm 1.45$ | $85.93 \pm 0.24$ |
| LELP (Ours) | $89.22 \pm 0.06$ | $\mathbf{81.43 \pm 0.47}$ | $92.81 \pm 0.36$ | $\mathbf{88.49 \pm 0.36}$ | $\mathbf{91.76 \pm 0.17}$ | $\mathbf{78.08 \pm 0.81}$ | $\mathbf{77.48 \pm 0.43}$ | $\mathbf{87.00 \pm 0.25}$ |
| Avg. gain over the best baseline | −0.02 | 0.20 | −0.04 | 1.15 | 0.36 | 1.85 | 0.21 | 0.88 |
| Avg. gain over non-subclass baseline | 0.24 | 0.20 | 0.08 | 1.73 | 0.36 | 2.54 | 1.68 | 0.88 |
| Avg. gain over Vanilla KD | 0.52 | 0.93 | 0.42 | 1.73 | 3.16 | 2.95 | 5.18 | 1.18 |

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

## A BROADER IMPACT STATEMENT

Due to its popularity and the fact that deep learning is widely used in fields from NLP to robotics to autonomous vehicles, knowledge distillation, as a deep learning method, carries with it the potential for harmful societal impact. However, we feel that none of these impacts must be specifically highlighted here.

## B BINARY CLASSIFICATION WITH SUBCLASS STRUCTURE: EXPERIMENTAL RESULTS

Here we present the table with the experimental results corresponding to Sections 4.2 and 4.3.1.

Table 3: Experiments on Binary Classification tasks **with** Subclass Structure.

| Teacher Architecture | ResNet92 | ResNet92 | MobileNetWD2 | ResNet92 | ResNet92 | MobileNetWD2 |
|---|---|---|---|---|---|---|
| Student Architecture | ResNet56 | MobileNet | MobileNet | ResNet56 | MobileNet | MobileNet |
| Dataset | CIFAR-10-bin | CIFAR-10-bin | CIFAR-10-bin | CIFAR-100-bin | CIFAR-100-bin | CIFAR-100-bin |
| Teacher | 96.70 | 96.70 | 93.90 | 78.69 | 78.69 | 72.63 |
| Subclass Distillation Teacher | 96.12 | 95.99 | 93.82 | 77.52 | 75.87 | 71.53 |
| Oracle Clustering | $97.17 \pm 0.02$ | $93.48 \pm 0.05$ | $93.48 \pm 0.05$ | $86.59 \pm 0.05$ | $86.38 \pm 0.06$ | $86.38 \pm 0.06$ |
| Standard Training | $95.43 \pm 0.07$ | $92.04 \pm 0.26$ | $92.13 \pm 0.17$ | $74.80 \pm 0.52$ | $69.13 \pm 0.32$ | $68.84 \pm 0.97$ |
| Vanilla Distillation | $95.75 \pm 0.21$ | $92.91 \pm 0.17$ | $92.79 \pm 0.3$ | $72.87 \pm 0.27$ | $72.16 \pm 0.39$ | $71.45 \pm 0.59$ |
| Embedding Distillation | $96.48 \pm 0.1$ | $93.05 \pm 0.13$ | $92.57 \pm 0.06$ | $78.77 \pm 0.35$ | $72.24 \pm 0.20$ | $\underline{71.99 \pm 0.30}$ |
| FitNet | $\underline{96.53 \pm 0.05}$ | $93.31 \pm 0.11$ | $92.72 \pm 0.1$ | $79.14 \pm 0.18$ | $71.76 \pm 0.26$ | $68.79 \pm 2.82$ |
| VID | $96.13 \pm 0.16$ | $89.95 \pm 1.34$ | $84.09 \pm 3.38$ | $75.69 \pm 0.55$ | $71.99 \pm 0.18$ | $61.15 \pm 0.40$ |
| Relational KD | $96.22 \pm 0.09$ | $92.98 \pm 0.21$ | $\underline{92.81 \pm 0.14}$ | $78.57 \pm 0.64$ | $72.53 \pm 0.23$ | $71.66 \pm 0.1$ |
| DKD | $95.75 \pm 0.21$ | $92.91 \pm 0.17$ | $92.79 \pm 0.3$ | $72.87 \pm 0.27$ | $72.16 \pm 0.39$ | $71.45 \pm 0.59$ |
| CRD | $95.83 \pm 0.25$ | $92.47 \pm 0.13$ | $92.34 \pm 0.18$ | $73.01 \pm 0.21$ | $72.15 \pm 0.63$ | $69.53 \pm 1.37$ |
| Subclass Distillation | $96.44 \pm 0.06$ | $\underline{93.76 \pm 0.06}$ | $92.77 \pm 0.15$ | $\underline{79.23 \pm 0.17}$ | $\underline{73.90 \pm 0.18}$ | $70.60 \pm 0.27$ |
| Agglomerative | $95.48 \pm 0.02$ | $92.16 \pm 0.15$ | $92.14 \pm 0.28$ | $76.84 \pm 0.38$ | $68.05 \pm 1.80$ | $66.97 \pm 0.93$ |
| K-means | $95.76 \pm 0.09$ | $92.53 \pm 0.32$ | $92.05 \pm 0.14$ | $77.03 \pm 0.21$ | $69.4 \pm 0.87$ | $68.97 \pm 0.09$ |
| t-SNE & K-means | $95.92 \pm 0.19$ | $92.97 \pm 0.07$ | $91.20 \pm 0.52$ | $77.39 \pm 0.29$ | $72.19 \pm 0.60$ | $68.15 \pm 0.09$ |
| LELP (Ours) | $\mathbf{96.71 \pm 0.04}$ | $\mathbf{93.99 \pm 0.17}$ | $\mathbf{93.03 \pm 0.1}$ | $\mathbf{79.91 \pm 0.15}$ | $\mathbf{75.21 \pm 0.12}$ | $\mathbf{72.38 \pm 0.24}$ |
| Avg. gain over the best baseline | 0.23 | 0.23 | 0.22 | 0.68 | 1.31 | 0.39 |
| Avg. gain over non-subclass baseline | 0.23 | 0.68 | 0.22 | 0.77 | 2.68 | 0.39 |
| Avg. gain over Vanilla KD | 0.96 | 1.08 | 0.24 | 7.04 | 3.05 | 0.93 |

## C ABLATIONS

One of the key design choices in LELP is to use the top PCA directions to construct the pseudo-subclasses. This is based on the intuition that axis which contain the most variation also contain the most information for distillation. Here we test this assumption by comparing PCA projections against three baselines. The first baseline is random projections (**Rand**), in which we choose $S$ random orthogonal directions to use as our subclass direction. The second baseline is raw PCA, which is using the top-K PCA directions, without the additional projection and random rotation applied, as we discussed in Section 3. This corresponds to using vectors $v_{c,s}$ instead of $\tilde{v}_{c,s}$. Our final baseline is the "Identity" projection, which is the same as using all of the teacher embeddings as subclasses. We consider the binarized CIFAR-100 task, and sweep $S$ from $2 - 256$, the embedding dimension of the teacher $D = 256$. We distill from ResNet92 to ResNet56, with the subclass temperature parameter $\beta$ ranging from $2^{-5} - 2^0$, with results shown in Figure 5 (top row), averaged over three runs. For a more clear comparison to the random projection baseline, we also show the "advantange" over random projections in fig. 5 (bottom row), made by subtracting the accuracy by the equivalent obtained by random projects with the same $S$ and $\beta$. Figure 5 shows the following general trends:

**Larger $S$ generally improves performance for Random Projections and LELP, and can harm PCA**. For Rand and LELP, performance seems mostly monotonically increasing with larger $S$. For LELP the benefit plateaus around $S = 32$, while for Rand, the performance slowly increases until $S = 256$. This suggests that the PCA step used in LELP obtains the salient information from teacher embeddings in fewer $S$ than Rand. For PCA without the extra steps used in LELP, large subclasses with higher temperatures significantly degrades performance, while adding rotation makes the performance of PCA stable. We conjecture that this is due to the first few PCA directions containing most of the variation and the remaining direction behaving as random noise.

**LELP consistently outperforms Random Projections**. We see in fig. 5, when compared to random projections, LELP almost always has a non-trivial advantage. The performance of Identity projections

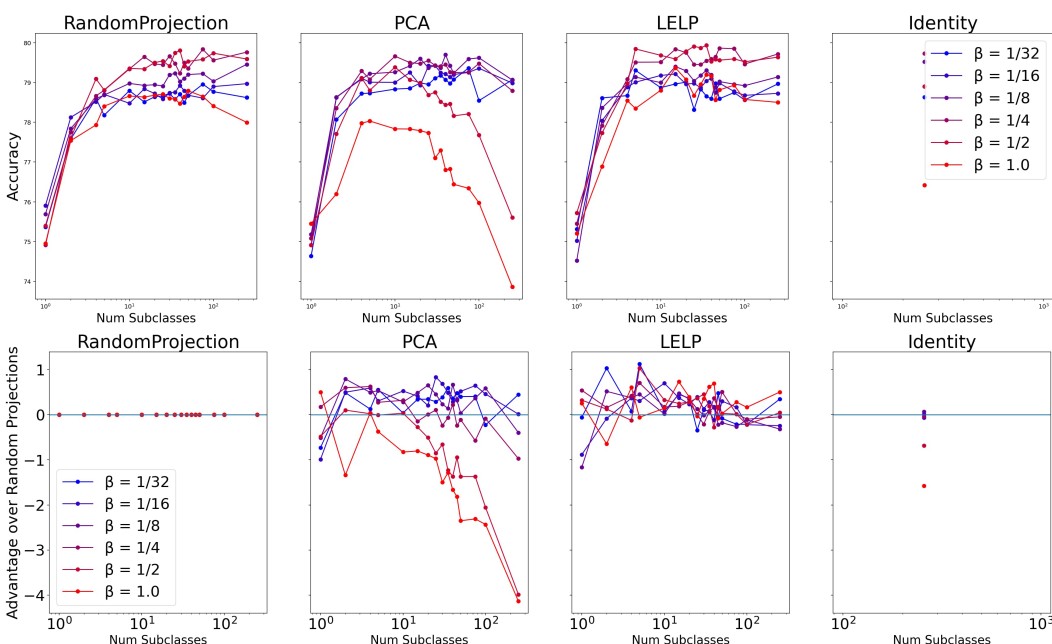

Figure 5: **Top row**: Ablations of choice of Projection, number of subclasses $S$ and subclass temperature $\beta$ on CIFAR-100bin. The set of plots displays raw CIFAR-100bin accuracy. **Bottom row**: The second set of plots demonstrates the accuracy gain achieved over random projections (using the same hyperparameter choice). Values over 0 indicate an advantage over random projections, which we see consistently with LELP.

is more inconsistent and it can be worse than random projections, depending on $\beta$. For larger number of subclasses, the advantage of LELP over random projections diminishes, as both methods now contain the majority of information from the entire embedding space.

# D    DETAILED DESCRIPTION OF THE NLP DATASETS WE USED.

The Corpus of Linguistic Acceptability (GLUE/cola) is a binary classification task which consists of English acceptability judgments drawn from books and journal articles on linguistic theory. Each example is a sequence of words annotated with whether it is a grammatical English sentence. GLUE/cola contains $8,551$ examples for training and $1,063$ examples for testing.

The Large Movie Review dataset (LMRD) is a dataset for binary sentiment classification containing $25,000$ movie reviews for training, and $25,000$ for testing.

The Stanford Sentiment Treebank (GLUE/sst2) consists of sentences from movie reviews and human annotations of their sentiment. The task is to predict the sentiment (positive or negative) of a given sentence. It contains $67,349$ examples for training and $1,821$ examples for testing.

The Amazon US reviews datasets comprise a vast collection of over a hundred million customer reviews and ratings (ranging from 1 to 5 stars). For our study, we selected a subset of these reviews across various products and devised two classification tasks: a 5-classes classification task and a binary classification task, which we describe below.

- "Amazon Reviews" contains $500,000$ examples for training and $10,000$ examples for testing. The task is to predict the star rating of the given review (ranging from 1 to 5). Both the training and test sets are balanced in terms of the number of examples per rating.

- "Amazon Reviews Bin" is a subset of "Amazon Reviews", where we exclude all 3-star reviews. It consists of $400,000$ training and $8,000$ testing examples. The objective is to determine whether a review is "polarized" (rated either 1 or 5 stars) or "mild" (2 or 4 stars).

For sampling the examples Amazon Reviews dataset we use the following process. We first first sequentially parse and concatenate the first 200k examples of the following datasets: "Office Products", "Video Games", "Video", "Toys", "Tools", "Sports", "Jewelry", "Digital Music Purchase", "Mobile Apps", "Video DVD", "Watches" (in this order). We then sequentially parse the resulting collection of examples and we add examples to the training and testing datasets. In particular, for each class (1-5 stars), we add the first 100k examples we encounter to the training dataset, and the next 2k examples we encounter to the test dataset.

The Sentiment140 dataset contains $1,600,000$ tweets extracted using the twitter api, which have been annotated either as positive or negative. We sample $1,590,000$ of these tweets to create a balanced training set, and we use the rest $10,000$ as the test set (also balanced). Note that the Sentiment140 datasets also comes with a validation set of $498$ examples, part of which is annotated as "Neutral". We are not making use of the latter validation set which is why in our tables we refer to the dataset we use (described earlier) as "Sentiment140 Bin".

# E  MULTICLASS CLASSIFICATION

While our primary focus lies on classification tasks with few classes, here we present results demonstrating the applicability of LELP to multiclass classification tasks with a moderate number of classes. To this end, we evaluate LELP on the CIFAR-10 and CIFAR-100 datasets. It is important to note that our objective is not to establish state-of-the-art performance in vision tasks but rather to showcase the versatility of our approach. We selected CIFAR-10 and CIFAR-100 as our datasets because they are widely recognized benchmarks for 10-class and 100-class classification, respectively. We intentionally disregarded image-specific techniques, such as data augmentation, to focus on a more general comparison. Consequently, we did not include methods like CRD and DKD that are specifically tailored to image data. Our results indicate that LELP can effectively handle tasks with a moderate number of classes and outperforms modality-independent knowledge distillation methods, particularly in scenarios where there is a significant disparity between the teacher and student architectures. As the number of classes increases, we anticipate LELP's performance to converge with vanilla knowledge distillation. Consequently, we did not conduct experiments on datasets with a large number of classes, such as ImageNet-1k.

Our results are shown in Table 4, where we distill a ResNet92 to a MobileNet. In this comparison, LELP significantly outperforms baselines, seeing a 1.1% improvement and 2.31% improvement over the next best baseline in CIFAR-10 and CIFAR-100, respectively. This validates the hypothesis that providing subclass information in the form of a classification loss avoids the pitfalls that methods which directly match embeddings have when students and teachers differ. When there is either a large performance or architecture gap, embeddings do not transfer readily. Appendix F.1 contains additional experiments in semi-supervised knowledge distillation, where we find that LELP consistently outperforms existing baselines without modification in the semi-supervised setting.

Table 4: Experiments on **Multi-class Classification tasks**.

| | | |
|---|---|---|
| Teacher Architecture | ResNet92 | ResNet92 |
| Student Architecture | MobileNet | MobileNet |
| Dataset | CIFAR-10 | CIFAR-100 |
| Teacher | 94.94 | 75.55 |
| Subclass Distillation Teacher | 93.49 | 70.23 |
| Standard Training | $84.86 \pm 0.11$ | $53.15 \pm 0.21$ |
| Vanilla Distillation | $86.45 \pm 0.21$ | $56.20 \pm 0.11$ |
| Embedding Distillation | $86.82 \pm 0.16$ | $57.27 \pm 0.4$ |
| FitNet | $85.70 \pm 0.36$ | $38.23 \pm 1.12$ |
| VID | $76.36 \pm 1.56$ | $27.08 \pm 1.08$ |
| Relational KD | $86.92 \pm 0.34$ | $59.67 \pm 0.58$ |
| Subclass Distillation | $86.55 \pm 0.28$ | $54.45 \pm 1.25$ |
| LELP (Ours) | $\mathbf{88.02 \pm 0.19}$ | $\mathbf{61.98 \pm 0.08}$ |
| Avg. gain over the best baseline | 1.1 | 2.31 |
| Avg. gain over non-subclass baseline | 1.1 | 2.31 |
| Avg. gain over Vanilla KD | 1.57 | 5.78 |

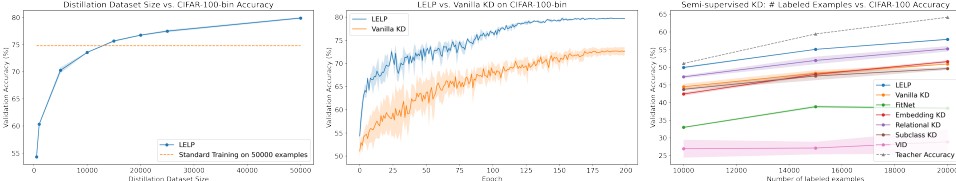

Figure 6: Experiments on the binary and standard CIFAR-100 datasets using ResNet92 as the teacher and ResNet56 and MobileNet, respectively, as the student. **Left:** Distillation Dataset Size vs. Accuracy on binary CIFAR-100. LELP achieves the same performance as standard training while using only 25% of the data. **Middle:** Student's validation accuracy over the training trajectory. LELP offers both performance gains over Vanilla KD and a faster convergence rate. **Right:** Illustration of the performance of LELP in a semi-supervised setting. The x-axis shows the initial quantity of labeled examples used to train the teacher model, which then generates pseudo-labels for the remaining (unlabeled) portion of the CIFAR-100 dataset. See Appendix F.1 for more details.

# F DATA EFFICIENCY, TRAINING SPEED AND ROBUSTNESS

As we demonstrate in Figure 6, LELP comes with several desirable properties. We consider the binary and standard CIFAR-100 datasets, ResNet92 as the teacher and ResNet56 and MobileNet, respectively, as the student. As we have already discussed in Appendix E, we selected CIFAR-10 and CIFAR-100 as our datasets because they are widely recognized benchmarks for 10-class and 100-class classification, respectively, and we intentionally disregarded methods that are tailored to vision-tasks.

We observe the following. First, the teacher conveys a high amount of information per example, in the sense that LELP is able to achieve the same performance as standard training using only a small fraction of the data. Second, LELP both outperforms and converges faster than Vanilla KD. Finally, in Appendix F.1 we show that LELP can offer significant gains in the semi-supervised setting, where the teacher is likely to generate inaccurate pseudo-labels.

## F.1 SEMI-SUPERVISED KD EXPERIMENTS

Table 5: Experiments in the **semi-supervised KD setting.**

| | | | |
|---|---|---|---|
| Teacher Architecture | ResNet92 | ResNet92 | ResNet92 |
| Student Architecture | MobileNet | MobileNet | MobileNet |
| Dataset | CIFAR-100 | CIFAR-100 | CIFAR-100 |
| Number of Labeled Examples | 10000 | 15000 | 20000 |
| Teacher | 51.08 | 59.43 | 64.16 |
| Subclass Distillation Teacher | 53.05 | 60.22 | 63.93 |
| Standard Training | $44.49 \pm 0.82$ | $48.31 \pm 0.90$ | $50.92 \pm 0.04$ |
| Vanilla Distillation | $44.49 \pm 0.82$ | $48.31 \pm 0.90$ | $50.92 \pm 0.04$ |
| Embedding Distillation | $42.46 \pm 0.47$ | $48.01 \pm 0.35$ | $51.60 \pm 0.38$ |
| FitNet | $33.0 \pm 0.21$ | $38.85 \pm 0.31$ | $38.43 \pm 0.29$ |
| VID | $26.97 \pm 2.53$ | $27.16 \pm 1.78$ | $28.91 \pm 3.38$ |
| Relational KD | $47.29 \pm 0.38$ | $51.94 \pm 1.06$ | $55.22 \pm 0.76$ |
| Subclass Distillation | $43.77 \pm 0.38$ | $47.53 \pm 1.17$ | $49.64 \pm 0.38$ |
| LELP (Ours) | $\mathbf{49.95 \pm 0.22}$ | $\mathbf{55.07 \pm 0.09}$ | $\mathbf{57.90 \pm 0.16}$ |
| Avg. gain over the best baseline | 2.66 | 3.13 | 2.68 |
| Avg. gain over non-subclass baseline | 2.66 | 3.13 | 2.68 |
| Avg. gain over Vanilla KD | 5.46 | 6.76 | 6.98 |

Semi-supervised KD, also known as KD with unlabeled examples, is a potent training paradigm for generating compact and efficient student models in scenarios where labeled data is scarce but a large pool of unlabeled data exists. This approach employs a high-capacity teacher model to generate (soft) pseudo-labels for the unlabeled dataset, which are subsequently utilized to train the student model.

Despite its widespread success in practice, the effectiveness of this powerful approach generally depends on the quality of the pseudo-labels generated by the teacher model. Indeed, training the student model on noisy pseudo-labels often leads to significant degradation of its generalization performance, and this is a well-known phenomenon that has been observed and studied in a plethora

of papers in the literature, e.g., Arazo et al. (2020); Liu & Tan (2021); Pham et al. (2021); Stanton et al. (2021); Xie et al. (2020); Baykal et al. (2023); Iliopoulos et al. (2022); Kontonis et al. (2023). Additionally, enforcing more teacher-student consistency, e.g., by blindly mimicking the teacher's embeddings, can even hinder performance when the teacher's output contains high noise (i.e., one may get worse performance than simply applying Vanilla KD). Finally, in this setting, often times the expense of using the teacher model to generate pseudo-labels can be a significant limitation. This becomes particularly problematic for methods like Subclass Distillation, which necessitate training multiple teacher models and generating pseudo-labels with each one of them, further compounding the cost.

In Table 5 (see also the rightmost plot in Figure 6) we study such a semi-supervised setting in the case of CIFAR-100, with the teacher model being a ResNet92 and student model being a MobileNet. (We choose this setting because it is the task where the teacher models will be the most noisy, and also the teacher and student models have different architectures.)

We consider the cases where the teacher model has available 10000, 15000, 20000 labeled examples for training, and it is then used to pseudo-label the rest of the CIFAR-100 dataset. (Note that the amount of available labeled examples for training the teacher model directly affects its accuracy.) Finally, the student-model is trained on both the available labeled data (which are also soft-labeled by the teacher model) and the teacher's pseudo-labels on the unlabeled data.

We observe that LELP provides significant gains in this setting. Also notably, for the 10000 and the 15000 case, the only baseline that outperforms Vanilla KD is Relational KD — showing that, while the teacher's embeddings in a noisy teacher setting may contain valuable information, extracting it effectively is a complex challenge — especially when the teacher and student come from different architecture families.

# G   DETAILED DESCRIPTION OF LELP

Here we provide further details and pseudo-code for the steps detailed in Section 3.

## G.1   STEP 1: PCA ON TEACHER EMBEDDINGS

---

**Algorithm 1** Learning Embedding Linear Projections (LELP) - Step 1 - Computing Subclass Direction

**Input:** Teacher model feature extractor $h^{\text{Teacher}}(x)$, teacher final layer weight $W_T \in R^{D \times C}$, dataset with class labels $\{x_i, y_i\}_{i \leq N}$, with class counts $N_c$
**Output:** $S \times C$ class vectors $\tilde{V} = \{\tilde{v}_{c,s}\}$ and $C$ class means $M = \{\mu_c\}$

---

Instantiate set of $\tilde{v}_{c,s}$ as empty set $\tilde{V} \leftarrow \{\}$
Compute QR Decomposition of $W_T$, $\tilde{W}_T \leftarrow \text{QR}(W_T)$ with Gram-Schmidt Algorithm
**for** $c = 1$ **to** $C$ **do**
    Get set of all inputs $X^c = \{x_i\}$ belonging to class $c$
    Compute matrix of teacher features for class $c$, $H^c \in R^{N_c \times D}$, s.t. $H_i^c = h^{\text{Teacher}}(X_i^c)$
    Project $H^c$ onto null space of $W_T$: $\tilde{H}^c \leftarrow H^c - H^c \tilde{W}_T$
    Compute top-$S$ PCA on $\tilde{H}^c$ to obtain $\{v_{c,s}\}$ for $1 \leq s \leq S$
    Produce random orthogonal matrix $Q^c \in R^{S \times S}$
    Set $\tilde{v}_{c,s} \leftarrow Q^c v_{c,s}$ for $1 \leq s \leq S$
    Compute $\{\sigma_{c,s}\}$, as $\sigma_{c,s}^2 = \frac{1}{N_c} \|\tilde{H}^c \tilde{v}_{c,s}\|_2^2$
    Normalize $\tilde{v}_{c,s} \leftarrow \frac{\tilde{v}_{c,s}}{max_s\{\sigma_{c,s}\}}$ (Normalizing $\tilde{v}_{c,s}$)
    Add $\{\tilde{v}_{c,s}\}$ to $\tilde{V}$: $\tilde{V} \leftarrow \tilde{V} \cup \{\tilde{v}_{c,s}\}$
**end for**
**Return:** $\tilde{V}$

---

This corresponds to section 3.1, and the main goal of this step is to obtain $S \times C$ vectors $\tilde{v}_{s,c} \in R^D$ to create subclasses from, with $D$ the teacher embedding dimension. The pseudocode is provided in algorithm 1. Note that the version provided in algorithm 1 works with the full teacher embedding

matrix for a class $H^c \in R^{N_c \times D}$, but it is straightforward to adapt this to a streaming approach that does not require storing all the embeddings at the same time by keeping running statistics. Note we additionally perform a small normalization step, so that the maximum variance along any of the teacher projection vectors $\{\tilde{v}_{c,s}\}$ is 1 for any class.

## G.2 STEP 2/3: KNOWLEDGE DISTILLATION WITH SUBCLASSES

---

**Algorithm 2** Subclass Splitting from teacher embedding, $\texttt{subsplit}(h, \tilde{V}, M, W, \beta, \tau)$

---

**Input:** Teacher embedding $h$
      $C \times S$ subclass projection vectors $\tilde{V} = \{\tilde{v}_{c,s}\}$
      $C$ class means $M = \{\mu_c\}$
      Teacher final layer classification weights $W = [w_1 \dots w_C] \in R^{D \times C}$ and biases $[b_1 \dots b_C]$
      Subclass Temperature $\beta$
      Student-Teacher Temperature $\tau$
**Output:** $C \times S$ Teacher subclass probabilities: $p_{c,s}^{\text{Teacher}}$

---

Compute teacher coarse label logits: $z_c \leftarrow w_c^\mathsf{T} h - b_c$
Compute teacher $C$ coarse label probabilities with temperature $\tau$: $p_c^{\text{Teacher}} \leftarrow \frac{e^{z_c/\tau}}{\sum_{j=1}^{C} e^{z_c/\tau}}$ ($\tau$-tempered Softmax)
**for** $c = 1$ **to** $C$ **do**
    Compute $S$ subclass logits: $z_{c,s} \leftarrow \tilde{v}_{c,s}^\mathsf{T}(h - \mu_c)$
    Compute subclass probabilities for class c: $p_{c,s}^{\text{Teacher}} \leftarrow p_c^{\text{Teacher}} * \frac{e^{z_{c,s}/\beta}}{\sum_{j=1}^{S} e^{z_{c,j}/\beta}}$. ($\beta$-tempered softmax over subclass logits)
**end for**
**Return:** $p_{c,s}^{\text{Teacher}}$

---

---

**Algorithm 3** Learning Embedding Linear Projections (LELP) - Knowledge Distillation with Subclasses

---

**Input:** Teacher model feature extractor $h^{\text{Teacher}}(x)$
      $C \times S$ subclass projection vectors $\tilde{V} = \{\tilde{v}_{c,s}\}$
      $C$ class mean vectors $M = \{\mu_c\}$
      Teacher final layer classification weights $W = [w_1 \dots w_C] \in R^{D \times C}$ and biases $[b_1 \dots b_C]$
      Subclass Temperature $\beta$
      Student-Teacher Temperature $\tau$
      $C \times S$ class student model $f_{\theta_S}^{\text{Student}}(x)$ and weights $\theta_S$
      Dataset $p(x)$
      Learning rate $\eta$
**Output:** Trained Student Model $\theta$
**while** Not converged **do**
    Sample $x \sim p(x)$
    Compute teacher embedding $h^{\text{Teacher}} \leftarrow h^{\text{Teacher}}(x)$
    Compute teacher subclass probabilities $p_{s,c}^{\text{Teacher}} \leftarrow \texttt{subsplit}(h, \tilde{V}, M, W, \beta, \tau)$
    Compute $C \times S$ student logits: $z_{c,s}^{\text{Student}} \leftarrow f_{\theta_S}^{\text{Student}}(x)$
    Compute $C \times S$ tempered student probabilities: $p_{c,s}^{\text{Student}} \leftarrow \frac{e^{z_{c,s}^{\text{Student}}/\tau}}{\sum_{i=1}^{C} \sum_{j=1}^{S} e^{z_{i,j}^{\text{Student}}/\tau}}$
    Compute LELP loss $\mathcal{L}_{LELP} = \tau^2 \sum_{c=1}^{C} \sum_{s=1}^{S} p_{c,s}^{\text{Teacher}} \log \frac{p_{c,s}^{\text{Teacher}}}{p_{c,s}^{\text{Student}}}$ (Standard KL Divergence)
    Update student parameters: $\theta_S \leftarrow \theta_S - \eta \frac{\partial \mathcal{L}_{LELP}}{\partial \theta_S}$
**end while**
**Return:** $\theta_S$

---

This corresponds to Section 3.2 and Section 3.3. In Algorithm 2 we describe how to generate the pseudo-subclasses from a given teacher embedding, $h$. This subprocess is used in Algorithm 3 for

knowledge distillation. In Algorithm 3 we describe the pure-knowledge distillation setting (i.e. with $\alpha = 0$ in eq. (1)), meaning we are not using the hard labels at all, as we did in the main text, but it is straightforward to combine with with the standard cross-entropy loss with hard labels using eq. (1).

# H  IMPLEMENTATION DETAILS

We implemented all algorithms in Python and used the TensorFlow deep learning library Abadi et al. (2016). We ran our experiments on 64 Cloud TPU v4s each with two cores.

For a fair comparison, we use the teacher's last-layer embeddings throughout the distillation process across all relevant baselines (Embedding Distillation, FitNet, VID, Relational KD, CRD). Furthermore, to handle mismatches in embedding dimensions between teacher and student, we introduce a trainable fully connected layer as a learnable projection for all these methods. (For Vision datasets, we have experimented with convolutional layers as suggested in Romero et al. (2014), and while these do reduce the size of the student model, they do not significantly impact its performance compared to the fully connected ones.)

For FitNet and CRD, in any given experiment, we pre-train the learnable projection once. This same pre-trained projection is then utilized consistently across all three trials corresponding to the experiment. The optimizer used for the pre-training of the learnable projection is Adam with initial learning rate $10^{-3}$ and $10^{-6}$ for Vision (200 epochs training) and Natural Language (40 epochs training) datasets, respectively.

We implement VID by using the loss function as described in (5) of Ahn et al. (2019) where the squared difference in the second term is taken over the teacher's and student's embeddings (using a learnable projection if there is a mismatch in their dimensions).

We implement Relational KD using the loss function as described in (9) and (10) of Park et al. (2019).

For CRD we implement the objective described in (17) of Tian et al. (2019) and perform grid search over $\{0.1, 1.0, 5.0, 10.0, 100.0\}$ for its coefficient. We consider the standard negative sampling policy where a pair of examples $(x, y)$ is considered "negative" if $x \neq y$, i.e., given a batch the number of positive pairs is equal to the batch size. (Note that in Vision classification tasks the standard approach would be to generate variations of every image (e.g, via rotations), and then also consider as positive any pair of examples (original or modified) that come from the same source. However, data augmentation of Natural Language examples is not straightforward and this impairs the performance of the method — this is the main point we are trying to convey here.)

For DKD we implement equations (1), (2) and (7) in Zhao et al. (2022). We set the 'target' hyperparameter $\alpha$ equal to 1 and perform grid search for the "non-target" hyperparameter $\beta$ over $\{1, 2, 4, 6, 8, 10\}$ (following examples in Zhao et al. (2022)).

The hyperparameters chosen for each method and dataset can be found in Appendix H.3.

## H.1  VISION DATASETS

In every experiment, both the teacher and student models are trained for 200 epochs.

For training ResNet-92 we use the SGD optimizer with initial learning rate $10^{-3}$, non-Nesterov momentum value equal to 0.9, cosine annealing learning rate schedule (minimum learning rate value is set to $10^{-6}$), and batch size 256. For training ResNet-56 we use the SGD optimizer with initial learning rate $5 \cdot 10^{-2}$, Nesterov momentum value equal to 0.9, cosine annealing learning rate schedule (minimum learning rate value is set to $10^{-6}$) and batch size 256. (This is a training schedule similar to the one described in Stanton et al. (2021)).

For training MobileNet (both the larger teacher-model and the student-model) we use the Adam optimizer with initial learning rate $\mathrm{lr} = 0.001$ and batch size 128. We then proceed according to the

following learning rate schedule for 200 epochs (see, e.g., He et al. (2016)):

$$lr \leftarrow \begin{cases} lr \cdot 0.5 \cdot 10^{-3}, & \text{if } \#\text{epochs} > 180 \\ lr \cdot 10^{-3}, & \text{if } \#\text{epochs} > 160 \\ lr \cdot 10^{-2}, & \text{if } \#\text{epochs} > 120 \\ lr \cdot 10^{-1}, & \text{if } \#\text{epochs} > 80 \end{cases}$$

Finally, in all cases we use data-augmentation. In particular, we use random horizontal flipping and random width and height translations with width and height factor, respectively, equal to $0.1$.

## H.2 NATURAL LANGUAGE DATASETS

For the GLUE/COLA dataset, the teacher model undergoes training for 2 epochs. In the case of GLUE/SST-2 and the Large Movie Review Dataset, teacher model training extends to 3 epochs. We employ the Adam optimizer (batch size $64$, initial learning rate $10^{-5}$) for these training processes. To minimize variance across experiments, we consistently train the student model for $40$ epochs using a smaller learning rate. The training uses the Adam optimizer with a batch size of $64$ and an initial learning rate of $10^{-6}$.

For both the Amazon Reviews datasets, the teacher and student models (both the ALBERT-base and the MLPs over frozen-T5 embeddings) are trained for 2 epochs using the Adam optimizer with a batch size of $64$ and an initial learning rate of $10^{-6}$.

For the Sentiment140 Bin dataset the teacher and the student models are trained for $1$ epoch using the Adam optimizer with batch size of $64$ and an initial learning rate of $10^{-6}$.

## H.3 HYPERPARAMETER OPTIMIZATION

In this section, we present the hyperparameter optimization (grid search) procedure we followed for each method, dataset and experiment of Section 4 and Appendix 4.2. For Vision datasets, **bold** numbers correspond to the hyperparameters chosen for the case of ResNet56 student, the number in *italicized* numbers correspond to the hyperparameters chosen for the case of MobileNet student with ResNet92 as a teacher and, finally, the underlined numbers correspond to the case of MobileNet student with (a larger) MobileNet as a teacher. For Natural Language datasets, **bold** numbers correspond to the case of ALBERT-base student, and *italicized* numbers correspond to the case of a MLP over frozen sentence-T5 (11B) embeddings as a student.

Table 6: Hyperparameters for Vanilla KD

| Dataset | Temperature |
|---------|-------------|
| CIFAR-10-bin | {**1.0**, 2.0, 3.0, 4.0, 5.0, *10.0*} |
| CIFAR-100-bin | {**1.0**, 2.0, 3.0, 4.0, 5.0, *10.0*} |
| CIFAR-10 | {1.0, **2.0**, 3.0, 4.0, *5.0*, 10.0}} |
| CIFAR-100 | {1.0, 2.0, 3.0, 4.0, 5.0, ***10.0***} |
| LMRD | {**1.0**, 2.0, 3.0, 4.0, 5.0, 10.0} |
| GLUE/cola | {1.0, 2.0, 3.0, **4.0**, 5.0, 10.0} |
| GLUE/sst2 | {1.0, 2.0, **3.0**, 4.0, 5.0, 10.0} |
| Amazon Reviews Bin | {*1.0*, 2.0, 3.0, **4.0**, 5.0, 10.0} |
| Amazon Reviews | {**1.0**, *2.0*, 3.0, 4.0, 5.0, 10.0} |
| Sentiment140 Bin | {**1.0**, 2.0, 3.0, 4.0, 5.0, 10.0} |

Table 7: Hyperparameters for Embedding KD

| Dataset | Temperature | Embeddings-loss coefficient |
|---------|-------------|----------------------------|
| CIFAR-10-bin | {**1.0**, 2.0, 3.0, 4.0, 5.0, *10.0*} | {0.1, *1.0*, 5.0, 10.0, **100.0**, 1000.0} |
| CIFAR-100-bin | {**1.0**, 2.0, 3.0, 4.0, 5.0, *10.0*} | {*0.1*, 1.0, 5.0, 10.0, **100.0**, 1000.0} |
| CIFAR-10 | {1.0, **2.0**, 3.0, 4.0, 5.0, *10.0*} | {*0.1*, 1.0, 5.0, 10.0, **100.0**, 1000.0} |
| CIFAR-100 | {**1.0**, 2.0, 3.0, 4.0, 5.0, *10.0*} | {*0.1*, **1.0**, 5.0, 10.0, 100.0, 1000.0} |
| LMRD | {1.0, 2.0, 3.0, **4.0**, 5.0, 10.0} | {**0.1**, 1.0, 5.0, 10.0, 100.0, 1000.0} |
| GLUE/cola | {1.0, **2.0**, 3.0, 4.0, 5.0, 10.0} | {0.1, **1.0**, 5.0, 10.0, 100.0, 1000.0} |
| GLUE/sst | {1.0, 2.0, **3.0**, 4.0, 5.0, 10.0} | {**0.1**, 1.0, 5.0, 10.0, 100.0, 1000.0} |
| Amazon Reviews Bin | {1.0, 2.0, 3.0, ***4.0***, 5.0, 10.0} | {*0.1*, 1.0, 5.0, 10.0, **100.0**, 1000.0} |
| Amazon Reviews | {*1.0*, 2.0, 3.0, 4.0, 5.0, 10.0} | {*0.1*, 1.0, **5.0**, 10.0, 100.0, 1000.0} |
| Sentiment140 Bin | {1.0, 2.0, **3.0**, 4.0, 5.0, 10.0} | {**0.1**, 1.0, 5.0, 10.0, 100.0, 1000.0} |

Table 8: Hyperparameters for FitNet

| Dataset | Temperature | Embeddings-loss coefficient |
|---------|-------------|----------------------------|
| CIFAR-10-bin | {**1.0**, 2.0, 3.0, 4.0, *5.0*, 10.0} | {0.1, 1.0, 5.0, 10.0, **100.0**, *1000.0*} |
| CIFAR-100-bin | {**1.0**, 2.0, 3.0, 4.0, 5.0, *10.0*} | {0.1, *1.0*, 5.0, 10.0, **100.0**, 1000.0} |
| CIFAR-10 | {**1.0**, 2.0, 3.0, *4.0*, 5.0, 10.0} | {0.1, 1.0, *5.0*, 10.0, **100.0**, 1000.0} |
| CIFAR-100 | {**1.0**, 2.0, 3.0, 4.0, *5.0*, 10.0} | {*0.1*, **1.0**, 5.0, 10.0, 100.0, 1000.0} |
| LMRD | {**1.0**, 2.0, 3.0, 4.0, 5.0, 10.0} | {0.1, **1.0**, 5.0, 10.0, 100.0, 1000.0} |
| GLEU/cola | {1.0, 2.0, 3.0, 4.0, **5.0**, 10.0} | {0.1, 1.0, 5.0, **10.0**, 100.0, 1000.0} |
| GLEU/sst | {1.0, 2.0, **3.0**, 4.0, 5.0, 10.0} | {0.1, 1.0, **5.0**, 10.0, 100.0, 1000.0} |
| Amazon Reviews Bin | {1.0, **2.0**, 3.0, 4.0, 5.0, *10.0*} | {0.1, 1.0, 5.0, *10.0*, 100.0, 1000.0} |
| Amazon Reviews | {1.0, 2.0, **3.0**, 4.0, *5.0*, 10.0} | {*0.1*, 1.0, 5.0, **10.0**, 100.0, *1000.0*} |
| Sentiment140 Bin | {1.0, 2.0, **3.0**, 4.0, 5.0, 10.0} | {**0.1**, 1.0, 5.0, 10.0, 100.0, 1000.0} |

Table 9: Hyperparameters for VID

| Dataset | Temperature | Embeddings-loss coefficient |
|---|---|---|
| CIFAR-10-bin | {**1.0**, 2.0, 3.0, 4.0, 5.0, *10.0*} | {*0.1*, **0.2**, 0.5, 1.0, 5.0, 10.0, 100.0} |
| CIFAR-100-bin | {1.0, **2.0**, 3.0, 4.0, *5.0*, 10.0} | {**0.1**, 0.2, *0.5*, 1.0, 5.0, 10.0, 100.0} |
| CIFAR-10 | {1.0, 2.0, 3.0, **4.0**, 5.0, *10.0*} | {***0.1***, 0.2, 0.5, 1.0, 5.0, 10.0, 100.0} |
| CIFAR-100 | {1.0, 2.0, *3.0*, 4.0, 5.0, **10.0**} | {**0.1**, 0.2, 0.5, *1.0*, 5.0, 10.0, 100.0} |
| LMRD | {1.0, 2.0, 3.0, 4.0, 5.0, **10.0**} | {**0.1**, 0.2, 0.5, 1.0, 5.0, 10.0, 100.0} |
| GLEU/cola | {1.0, **3.0**, 3.0, 4.0, 5.0, 10.0} | {**0.1**, 0.2, 0.5, 1.0, 5.0, 10.0, 100.0} |
| GLEU/sst | {1.0, 2.0, 3.0, 4.0, 5.0, **10.0**} | {**0.1**, 0.2, 0.5, 1.0, 5.0, 10.0, 100.0} |
| Amazon Reviews Bin | {1.0, 2.0, 3.0, *4.0*, 5.0, **10.0**} | {**0.1**, 0.2, 0.5, 1.0, 5.0, 10.0, *100.0*} |
| Amazon Reviews | {1.0, 2.0, 3.0, *4.0*, 5.0, **10.0**} | {**0.1**, 0.2, 0.5, 1.0, *5.0*, 10.0, 100.0} |
| Sentiment140 Bin | {**1.0**, 2.0, 3.0, 4.0, 5.0, 10.0} | {**0.1**, 1.0, 5.0, 10.0, 100.0, 1000.0} |

Table 10: Hyperparameters for Relational KD

| Dataset | Temperature | Embeddings-loss coefficient |
|---|---|---|
| CIFAR-10-bin | {1.0, **2.0**, 3.0, 4.0, 5.0, *10.0*} | {*0.1*, 0.2, 0.5, **1.0**, 5.0, 10.0, 100.0} |
| CIFAR-100-bin | {**1.0**, 2.0, 3.0, 4.0, 5.0, *10.0*} | {*0.1*, 0.2, 0.5, 1.0, 5.0, **10.0**, 100.0} |
| CIFAR-10 | {**1.0**, 2.0, 3.0, 4.0, 5.0, 10.0} | {0.1, 0.2, 0.5, 1.0, 5.0, 10.0, ***100.0***} |
| CIFAR-100 | {1.0, 2.0, 3.0, 4.0, 5.0, ***10.0***} | {**0.1**, 0.2, 0.5, 1.0, 5.0, 10.0, *100.0*} |
| LMRD | {**1.0**, 2.0, 3.0, 4.0, 5.0, 10.0} | {0.1, 0.2, 0.5, **1.0**, 5.0, 10.0, 100.0} |
| GLEU/cola | {1.0, **2.0**, 3.0, 4.0, 5.0, 10.0} | {**0.1**, 0.2, 0.5, 1.0, 5.0, 10.0, 100.0} |
| GLEU/sst | {**1.0**, 2.0, 3.0, 4.0, 5.0, 10.0} | {0.1, 0.2, 0.5, **1.0**, 5.0, 10.0, 100.0} |
| Amazon Reviews Bin | {**1.0**, 2.0, 3.0, *4.0*, 5.0, 10.0} | {0.1, 0.2, 0.5, 1.0, *5.0*, **10.0**, 100.0} |
| Amazon Reviews | {**1.0**, 2.0, 3.0, *4.0*, 5.0, 10.0} | {0.1, 0.2, 0.5, **1.0**, 5.0, *10.0*, 100.0} |
| Sentiment140 Bin | {**1.0**, 2.0, 3.0, 4.0, 5.0, 10.0} | {**0.1**, 1.0, 5.0, 10.0, 100.0, 1000.0} |

Table 11: Hyperparameters for Subclass Distillation

| Dataset | Num. Subclasses | Auxiliary loss Temp. | Auxiliary loss weight | Distill. Temp. |
|---|---|---|---|---|
| CIFAR-10-bin | {2, 3, 4, 5, **_10_**} | {1.0, **5.0**, _10.0_} | {0.0, **_0.1_**, 1.0, 5.0, 10.0, 100.0} | {1.0, **2.0**, *3.0*, 4.0, 5.0, 10.0} |
| CIFAR-100-bin | {5, 10, 15, 20, 25, 30, *35*, 40, 45, 50, **75**, 100} | {**1.0**, 5.0, *10.0*} | {**0.0**, 0.1, 1.0, 5.0, 10.0, 100.0} | {**1.0**, 2.0, 3.0, 4.0, 5.0, 10.0} |
| CIFAR-10 | {**2**, 4, 8, **10**} | {1.0, *5.0*, 10.0} | {0.0, **0.1**, 1.0, 5.0, 10.0, 100.0} | {1.0, **2.0**, *3.0*, 4.0, 5.0, 10.0} |
| CIFAR-100 | {**2**, 4, 8, 10} | {*1.0*, **5.0**, 10.0} | {0.0, 0.1, **1.0**, 5.0, *10.0*, 100.0} | {1.0, 2.0, ***3.0***, 4.0, 5.0, 10.0} |
| LMRD | {2, 3, 4, 5, **10**, 25, 50} | {1.0, **5.0**, 10.0} | {0.0, **0.1**, 1.0, 5.0, 10.0, 100.0} | {**1.0**, 2.0, 3.0, 4.0, 5.0, 10.0} |
| GLEU/cola | {2, 3, 4, 5, 10} | {1.0, 5.0, **10.0**} | {**0.0**, 0.1, 5.0, 10.0, 100.0} | {1.0, 2.0, 3.0, 4.0, 5.0, 10.0} |
| GLEU/sst2 | {2, 3, 4, 5, **10**} | {1.0, **5.0**, 10.0} | {0.0, **0.1**, 5.0, 10.0, 100.0} | {**1.0**, 2.0, 3.0, 4.0, 5.0, 10.0} |
| Amazon Reviews Bin | {2, **4**, 5, 10, 20, 40, 50, 100, 200, 500} | {*1.0*, 5.0, 10.0} | {**0.0**, 0.1, 5.0, 10.0, 100.0} | {*1.0*, **2.0**, 3.0, 4.0, 5.0, 10.0} |
| Amazon Reviews | {**2**, 4, 8, 16, 20, 40, 80, *200*} | {**1.0**, 5.0, 10.0} | {*0.0*, 0.1, **5.0**, 10.0, 100.0} | {*1.0*, **2.0**, 3.0, 4.0, 5.0, 10.0} |
| Sentiment140 Bin | {**2**, 4, 5, 10, 20, 40, 50, 100, 200, 500} | {**1.0**, 5.0, 10.0} | {**0.0**, 0.1, 5.0, 10.0, 100.0} | {**1.0**, 2.0, 3.0, 4.0, 5.0, 10.0} |

Table 12: Hyperparameters for LELP (ours)

| Dataset | Num. Subclasses | Subclass Temp. | Distill. Temp. |
|---|---|---|---|
| CIFAR-10-bin | {5, 10, **_20_**} | {1/32, *1/16*, 1/8, **1/4**, 1/2, 1} | {**1.0**, 2.0, *4.0*, 8.0, 10.0} |
| CIFAR-100-bin | {5, 10, 15, **20**, 25, 30, 35, 40, 45, *50*, 75, 100} | {1/32, 1/16, 1/8, *1/4*, **1/2**, 1} | {1.0, **2.0**, *4.0*, 8.0, 10.0} |
| CIFAR-10 | {2, 4, **8**, _10_} | {1/32, 1/16, 1/8, 1/4, **1/2**, 1} | {1.0, 2.0, **4.0**, 8.0, *10.0*} |
| CIFAR-100 | {**2**, 4, 8, 10} | {1/32, 1/16, 1/8, 1/4, 1/2, ***1***} | {1.0, 2.0, 4.0, **8.0**, *10.0*} |
| LMRD | {5, **10**, 15, 20} | {1/32, 1/16, 1/8, 1/4, 1/2, 1} | {**1.0**, 2.0, 3.0, 4.0, 5.0, 10.0} |
| GLEU/cola | {5, 10, **15**, 20} | {1/32, 1/16, 1/8, **1/4**, 1/2, 1} | {1.0, 2.0, **3.0**, 4.0, 5.0, 10.0} |
| GLEU/sst2 | {**5**, 10, 15, 20} | {1/32, 1/16, 1/8, 1/4, **1/2**, 1} | {1.0, **2.0**, 3.0, 4.0, 5.0, 10.0} |
| Amazon Reviews Bin | {2, *4*, 5, 10, 20, 40, 50, **100**, 200, 500} | {1/32, **1/16**, 1/8, 1/4, 1/2, 1} | {1.0, 2.0, 3.0, 4.0, **5.0**, 10.0} |
| Amazon Reviews | {2, 4, 8, 16, 20, 40, *80*, **200**} | {1/32, 1/16, **1/8**, 1/4, 1/2, *1*} | {*1.0*, **2.0**, 3.0, 4.0, 5.0, 10.0} |
| Sentiment140 Bin | {2, 4, 5, 10, 20, 40, 50, 100, 200, **500**} | {1/32, 1/16, 1/8, 1/4, 1/2, **1**} | {1.0, **2.0**, 3.0, 4.0, 5.0, 10.0} |

Table 13: Hyperparameters for Agglomerative Clustering

| Dataset | Number of Clusters |
|---|---|
| CIFAR-10-bin | {**4**, 6, 8, *10*, 20, 50, 100, 1000} |
| CIFAR-100-bin | {**10**, 20, 30, 40, 50, 60, *70*, 80, 90, 100, 150, 200} |

Table 14: Hyperparameters for K-Means Clustering

| Dataset | Number of Clusters |
|---|---|
| CIFAR-10-bin | {4, *6*, 8, **10**, 20, 50, 100, 1000} |
| CIFAR-100-bin | {**10**, 20, 30, 40, *50*, 60, 70, 80, 90, 100, 150, 200} |

Table 15: Hyperparameters for t-SNE & K-Means Clustering

| Dataset | Number of Clusters |
|---|---|
| CIFAR-10-bin | {2, **3**, *4*, 5, 10, 25, 50, 500} |
| CIFAR-100-bin | {*5*, 10, 15, 20, 25, 30, 35, 40, 45, 50, 75, 100} |

