# OpenReview forum: "Linear Projections of Teacher Embeddings for Few-Class Distillation"
_ICLR.cc/2025/Conference — Submitted to ICLR 2025_

### Official Review · Reviewer_kvSt · 2024-10-22

**Soundness:** 3
**Presentation:** 2
**Contribution:** 3
**Rating:** 6
**Confidence:** 3

**Summary:**

This paper aims to improve the effectiveness of knowledge distillation on datasets with few classes. Specifically, the authors propose identifying the informative linear subspaces in the teacher’s embedding space and generating multiple pseudo labels from the perspective of different subspaces. In this way, the number of classes can be increased to improve the distillation performance on the few-class dataset. Experimental results and ablation studies demonstrated the effectiveness of the proposed method.

**Strengths:**

(1) The motivation is clear. The authors expand the number of classes by mapping the teacher’s embedding into different subspaces to generate pseudo labels.

(2) The proposed method fully explores the latent structure hidden in the teacher’s embedding and avoids re-training the teacher network for pseudo-label generation, which is interesting.

(3) The authors conduct extensive experiments under different settings, such as binary-class and few-class distillation, and on different datasets, such as Amazon Reviews and Sentiment140. As shown in the experiments, the proposed method outperforms the existing distillation methods by a clear margin.

**Weaknesses:**

(1) Since the main advantage of the proposed method over the Subclass Distillation is re-training free, it would be better to compare the training costs of different methods in the Experiments.

(2) As shown in Algorithm 1, the proposed method obtains the subclass direction by feeding the training samples into the teacher network. Since the data augmentation will be different at each training epoch, I am wondering if the effectiveness of the subclass direction is degraded by computing in advance.

(3) From Figure 5, it seems that random projections already achieve stable and promising performance. How about using an ensemble of random projections with different initializations to generate the subclasses?

(4) The compared distillation methods are not new. The latest method was published in 2022.

(5) There are many typos in the current manuscript. For instance, “form”->”from” in line 073, “they authors”->”the authors” in line 149. “coarse-graned”->”coarse-grained” in line 202. The authors should carefully check the whole manuscript.

**Questions:**

Please refer to the weaknesses.

---

> ### Author Response · Authors · 2024-11-19
> **Response to Reviewer  kvSt**
>
> We thank the reviewer for the time and effort put in evaluating our work. We have responded to the reviewer's comments below and are happy to answer any additional questions.
>
> **1. Since the main advantage of the proposed method over the Subclass Distillation is re-training free, it would be better to compare the training costs of different methods in the Experiments.**
>
> First, let us emphasize that our method significantly outperforms Subclass Distillation in most scenarios (e.g., in the largest datasets considered in the study: Amazon Reviews/ Sentiment140). In other words, the main advantage of our method is not just that it is re-training free.
>
> Second, as detailed in Lines 151-161 and Lines 225-247, these approaches often necessitate retraining the teacher multiple times for hyperparameter optimization. Given that Subclass Distillation has 4 hyperparameters, this makes the training process many orders of magnitudes more expensive (depending on the size of the teacher model). Even worse, if one is given only inference-access to the teacher model, as is often the case in real-world scenarios, Subclass Distillation is not even applicable.
>
>
> Since LELP and the other distillation methods explored in this study have negligible computational overhead, a detailed analysis of training costs falls outside the scope of this work. We believe such an analysis would detract from the core message of the paper.
>
> That said, we will make sure to emphasize the above facts in the next version of our paper, and provide explicit numbers and figures to illustrate the stark difference in training expenses within a specific scenario as an example.
>
>
>
> **2. As shown in Algorithm 1, the proposed method obtains the subclass direction by feeding the training samples into the teacher network. Since the data augmentation will be different at each training epoch, I am wondering if the effectiveness of the subclass direction is degraded by computing in advance.**
>
> Observe that when we create "augmented data", such as by rotating an image, the generated variations share the same label as the original. For example, if we rotate a picture of a cat multiple times, each rotated version is still labeled as a "cat". Consequently, a reliable teacher model should classify all these augmented images identically. This means the teacher model's 'embedding-vector' for these images should consistently represent a "cat". Since our pseudo-classes are derived from these embeddings, pre-computing them shouldn't lead to a performance drop.
>
>
> As a side remark, it's worth noting that our largest experiments involve NLP datasets, which don't typically use data augmentation. (However, this doesn't invalidate the point made earlier.)
>
> **3. From Figure 5, it seems that random projections already achieve stable and promising performance. How about using an ensemble of random projections with different initializations to generate the subclasses?**
>
> - We acknowledge the appeal of random projections due to their computational efficiency and conceptual simplicity. We extensively investigated their potential to match the performance of LELP and other clustering methods, recognizing the value of such an outcome. However, our experiments (see e.g. Figure 5, second row, third plot) revealed that LELP (and sometimes other clustering techniques) outperform random projections.
>
> - We are not really certain we have fully understood the details of what the reviewer is proposing (in the sense that our understanding is that averaging multiple random projections effectively results in another random projection, which does not provide additional information gain). Could the reviewer please elaborate on how such an ensemble might be constructed to improve performance ? —   we would be happy to conduct and report the results of the suggested experiment.
>
>
> **4. The compared distillation methods are not new. The latest method was published in 2022.**
>
> We have made every effort to include and compare against all relevant distillation methods applicable to our specific setting. However, we are always open to suggestions and would be happy to incorporate any additional methods that the reviewer deems relevant for a more comprehensive analysis.
>
>
> **5. There are many typos in the current manuscript. For instance, “form”->”from” in line 073, “they authors”->”the authors” in line 149. “coarse-graned”->”coarse-grained” in line 202. The authors should carefully check the whole manuscript.**
>
>
> Thank you — we will make sure to address these typos in the next version of our paper.

---

> > ### Comment · Reviewer_kvSt · 2024-11-25
> > **Response to Authors**
> >
> > Dear Authors,
> >
> > Many of my concerns are addressed properly. I am happy to increase one more score. However, there is no such option. Therefore, I will keep my initial rating.
> >
> > Thank You

---

> > > ### Author Response · Authors · 2024-11-25
> > >
> > > We thank the reviewer for their support and for acknowledging our responses!

---

### Official Review · Reviewer_VGkY · 2024-10-28

**Soundness:** 3
**Presentation:** 3
**Contribution:** 3
**Rating:** 6
**Confidence:** 3

**Summary:**

This paper introduces a knowledge distillation approach titled Learning Embedding Linear Projections (LELP) aimed at enhancing model performance in few-class classification tasks. LELP identifies informative linear subspaces within the teacher model's embedding space, splits them into pseudo-subclasses, and uses these to guide the training of the student model. Experimental results demonstrate that LELP outperforms existing state-of-the-art methods in large-scale NLP benchmarks such as Amazon Reviews and Sentiment140.

**Strengths:**

1. The author's viewpoint that "the information about the teacher model’s generalization patterns scales directly with the number of classes" is insightful and is not limited to knowledge distillation tasks.

2. The LELP method innovatively leverages structural information, i.e., "subclasses", in the teacher model's embedding space to enhance the performance of the student model, without the need to retrain the teacher model, and it is insensitive to differences in data types and model architectures.

**Weaknesses:**

1. Could you provide a more detailed explanation of why it is more effective to first project onto the "null-space" before performing PCA?

2. Why does random rotation guarantee that each direction has the same variance in expectation? Are there any theoretical insights regarding this?

3. When the number of categories in the dataset is sufficiently large, utilizing subclasses can further increase the category count. In this scenario, applying cross-entropy loss for distillation may weaken knowledge transfer for certain categories due to the additional reduction in gradient updates.

4. If the method is only effective with a very small number of categories, its generalizability is quite limited. The authors should include performance comparisons of additional model architectures on datasets with a greater number of categories, such as ImageNet, in Table 4.

5. The authors only provided the results of the grid search for hyperparameters without showing the corresponding test performance for different values. Therefore, I am unsure about the method's sensitivity to hyperparameters, and I believe this is very important for other researchers who wish to utilize this method.

**Questions:**

Please see the weaknesses.

---

> ### Author Response · Authors · 2024-11-19
> **Response to Reviewer VGkY**
>
> We thank the reviewer for the time and effort put in evaluating our work. We have responded to the reviewer's comments below and are happy to answer any additional questions.
>
> **1. [...] provide a more detailed explanation of why it is more effective to first project onto the "null-space" [...]**
>
>
> The motivation behind the null space projection is that the information along these directions is already contained in the original "super-class" soft-labels generated by the teacher, as we use  $w_1, … w_C$ to construct $p^{Teacher}_c$ (line 288). Therefore, we do not need this to encode this information again in the labels/directions of the pseudo-subclasses. Performing the null space projection before computing the PCA ensures that the subclass directions $v_1, … v_S$ are orthogonal to $w_1, … w_C$.
>
>
>
> **2. Why does random rotation [...] Are there any theoretical insights [...] ?**
>
>
> We assume that the reviewer is referring to the rotation operation given in lines 270-272. Suppose $V = [v_1, …. v_S]$ is our original projection vector. Let $Q$ be a random orthonormal matrix drawn from the Haar measure over the special orthogonal group $\mathrm{SO}(S)$ (i.e. rotationally uniform),  and assume $Q = [q_1, … q_S]$.
>
> Since the distribution of $Q$ is rotationally uniform, $q_1, … q_S$ have the same distribution. Furthermore, the covariance of our subclass projections after projecting onto $V$ is given by $\Sigma_V = \mathrm{diag}(\lambda_1, … \lambda_S)$, where $\lambda_s$ is the $s$-th largest eigenvalue of $\Sigma_c = \mathbb{E}[{h_c h_c^T}]$ (the embedding covariance). This is because $v_s$ is chosen as the $s$-th eigenvector of $ \Sigma_c$.
>
> Now after projecting onto $q_i$, this will have variance given by $q_i^T \Sigma_V q_i$. In expectation this has variance
>
> $\mathbb{E}[{q_i^T \Sigma_V q_i}] = \mathbb{E}[{ \mathrm{Tr}(q_i^T \Sigma_V q_i)}] =  \mathbb{E}[{ \mathrm{Tr}(q_i q_i^T \Sigma_V)}] = \mathrm{Tr}( \mathbb{E}[{ q_i q_i^T \Sigma_V}] )  =  \mathrm{Tr}( \frac{1}{S} I   \Sigma_V) = \frac{1}{S} \Sigma_{s = 1}^S \lambda_s  $,
> which is the same for all directions $q_i$.  Note that we make use of the fact that $\mathbb{E}[ q_i q_i^T] = \frac{I}{S} $ for each of $q_i$ when marginalized and $q_i$ is a random unit-norm vector in $R^S$.
>
> We will include the above to the next version of our paper.
>
> **3. When the number of categories in the dataset is sufficiently large [...]**
>
> As we mention in the Limations section of our work (Lines 497-506), we concur that LELP is not suitable for distillation in multi-class settings with a large number of classes as it is not designed for such scenarios. See also the General Response to all the reviewers.
>
>
> **4. [...] on datasets with a greater number of categories, such as ImageNet, in Table 4.**
>
> Our method is indeed designed for classification with a small number of classes, and we particularly focus on binary classification.
>
> We want to strongly emphasize that  binary classification is one of the most widespread and impactful classification types used today in real world scenarios, and this is what motivated our work. Its applications are far-reaching and crucial to the success of major multi-billion dollar industries. For example:
>  - **Click-through prediction:** A core component of online advertising, determining ad relevance and user engagement.
> - **Query understanding:** Essential for search engines to accurately interpret user intent and deliver meaningful results.
> - **Sentiment analysis:** A valuable tool for gauging public opinion, informing product development, and shaping marketing strategies.
>
>
>
> As acknowledged in the Limitations section (Section 5, Lines 497-506), our work does not address multi-class distillation with a large number of classes, such as ImageNet-1k. LELP's design specifically focuses on scenarios with fewer classes (this is because the necessity for supplementary pseudo-subclasses naturally decreases as the number of original classes grows). While our work focuses on few-class settings, we believe this strengthens, rather than diminishes, its significance. Our findings demonstrate that distillation techniques optimized for large-scale multi-class problems, such as ImageNet-1k, may underperform in the few-class setting, highlighting the need for different, potentially specialized, approaches.
>
>
>
>
> **5. The authors only provided the results of the grid search for hyperparameters [...]**
>
>
> We provided the results of the grid search for hyperparameters (and the values chosen) for reproducibility purposes — unfortunately though, we did not record the outcome of every single experiment conducted.
>
> That said, we have already conducted and presented ablations regarding how the hyperparameters affect the performance of our method  — see Figure 5 ( third column), and Appendix C for more details.
>
> We appreciate your suggestions and are happy to conduct further ablation studies as recommended to enhance the comprehensiveness of our work.

---

### Official Review · Reviewer_Hun6 · 2024-10-28

**Soundness:** 1
**Presentation:** 2
**Contribution:** 2
**Rating:** 5
**Confidence:** 5

**Summary:**

The main goal of the paper is to handle cases with relatively small number of classes when applying knowledge distillation. To this end, the authors enlarge the effective number of classes by projecting the final embedding vectors of the student and the teacher into several PCA subsets. This way, the teacher may contain more information that are distributed over many more effective classes, which transfer to the student during its training process.

**Strengths:**

- The problem that the paper deals with (specifically, handling small number of class in knowledge distillation) is important and valuable.
Also, the general idea of extending the clusters from the given classes to sub-classes is interesting and valid.
-The results that were reported are also somewhat encouraging.

**Weaknesses:**

- In section 3 the authors argued that in case the student and the teacher do not share the same embedding dimensions, a learnable projection layer is required which can often harm performance - however, the authors do not provide any explanation or evidence to this sentence (why it harms performance?) nor at least any reference to this determination. Also note that the proposed approach in the paper includes much more projection layers - why in this case the authors don’t think it can harm the performance?

- The authors proposed to use PCA to obtain the informative linear subspaces, which is an off-line process where only the final embedding layer was used. I am wondering whether these linear subspaces could be learned as part of the training of the teacher, to also output these additional embeddings? (e.g. using reconstruction loss)

- There is a significant effort to explain the setup in section 4.1 which I am wondering whether it was necessary, especially as the authors focus on cases where the teacher and student architectures have exactly the same dimensions which as I stated before, not sure why to limit to these cases?

- I would expect the authors to experiment also with regular (large) number of classes as CIFAR-100, TinyImageNet or other datasets to understand what are the limitations of the proposed approach and how it behaves on regular and common cases where there are many categories.

- I found it very hard to understand the t-SNE plots provided in Figure 4. What is the meaning of running different t-SNE for each one of the methods as each individual t-SNE run organizes the points differently? Why the shape of the embeddings look so different in the top row? Further explanation will be helpful.

- An intermediate analysis that shows the meaning of the sub-classes obtained by PCA  could help for visualization and understanding. For instance, would we observe meaningful fine-grained classes?

- Main concern is the weak experimental section. Only Table 2 provides detailed classification results. Only one teacher and student architectural choices were used. It is not clear how the method generalizes to other architectures. Also, the results are not convincing enough to my opinion and in some cases are marginal.

- What is the impact of the S hyper-parameter? (The number of sub-classes per class). I would expect some ablation study on this.

Minor:
Line 73: form —> from.
Line 202: coarse-graned —> coarse-grained

**Questions:**

I have already stated my questions in the 'weaknesses' section. Hope the authors can address my concern.

---

> ### Author Response · Authors · 2024-11-19
> **Response to Reviewer Hun6 (part 1)**
>
> We thank the reviewer for the time and effort put in evaluating our work. We have responded to the reviewer's comments below and are happy to answer any additional questions.
>
> - **In section 3 the authors argued that in case the student and the teacher do not share the same embedding dimensions, a learnable projection layer is required which can often harm performance - however, the authors do not provide any explanation or evidence to this sentence (why it harms performance?) nor at least any reference to this determination**
>
> We provide experimental evidence for the claim that learnable projections (i.e., the FitNet method) can oftentimes harm performance when the internal representations of the teacher and student have different sizes or come from fundamentally different network designs. See for example Lines 790-792 and:
>
> 1. Table 3, ResNet92 → MobileNet, CIFAR-100bin. Vanilla: 72.16%, FitNet: 71.76%
> 2. Table 3, MobileNetWD2 → MobileNet, CIFAR-100bin. Vanilla: 71.45%, FitNet: 68.79%
> 3. Table 4, ResNet92 → MobileNet, CIFAR-10. Vanilla: 86.45%, FitNet: 85.70%
> 4. Table 4, ResNet92 → MobileNet, CIFAR-100. Vanilla: 56.20%, FitNet: 38.23%
> 5. Let us also point out that in every experiment of Table 2 where the teacher and student have different embedding dimensions (i.e., in all but two of our  experiments), FitNet essentially is on par with Vanilla distillation, offering no real improvements.
>
>
> We will revise the relevant section to ensure this explanation is clearly conveyed to all readers
>
> - **Also note that the proposed approach in the paper includes much more projection layers - why in this case the authors don’t think it can harm the performance?**
>
> While our method indeed adds a layer to the student model, its function is distinct from the learnable projection in FitNet. Instead of simply copying the teacher's embeddings, our layer learns semantically meaningful relationships between pseudo-subclasses. This difference leads to superior performance in all our experiments, where our method consistently surpasses both FitNet and standard knowledge distillation.
>
> We will revise the relevant section to ensure this explanation is clearly conveyed to all readers.
>
>
> - **The authors proposed to use PCA to obtain the informative linear subspaces, which is an off-line process [...]**
>
> We appreciate the reviewer's perspective, but our goal is to explicitly avoid methods that require learning subspaces during teacher training (e.g. Subclass Distillation, which is similar in flavor to what the reviewer is proposing). As detailed in Lines 151-161 and Lines 225-247, these approaches often necessitate retraining the teacher multiple times for hyperparameter optimization, introducing impracticalities (for example, they rarely can be applied in an online fashion due to the hyperparameter optimization issue). Our focus is on developing techniques that operate efficiently without the need for repeated teacher retraining.
>
> That said, we recognize the benefits of online methods, but developing one is not within the scope of this current research.  However, we believe this is a promising avenue for future exploration.
>
>
> - **There is a significant effort to explain the setup in section 4.1 [...] the authors focus on cases where the teacher and student architectures have exactly the same dimensions?**
>
> On the contrary, in  Lines 346-356 we explain that our experiments **deliberately cover every case**, and do not focus on the same dimension case. These cases include scenarios where there is a mismatch in the teacher-student embedding dimensions, and scenarios with the family of the teacher and student architecture differ.
>
> We put the effort so that the reader is aware exactly that we do not limit ourselves to any particular scenario.
>
> We emphasize the following:
>
>  1. In our largest experiments (Amazon Reviews, Sentiment 140) which we highlight in Figure 1, the teacher and student embedding dimensions are indeed different. Even more, all but two of the experiments in Table 2 correspond to teacher-student pairs with different embedding dimensions.  Even for the two experiments where the embedding dimensions of the teacher and the student are equal, the underlying architectures are radically different (ALBERT-family, vs MLPs operating over T5 (11B) frozen embeddings.
>  2. Four out of six experiments in Table 3 consider teacher-student pairs with different embedding dimensions.

---

> ### Author Response · Authors · 2024-11-19
> **Response to Reviewer Hun6 (part 2)**
>
> -**I would expect the authors to experiment also with regular (large) number of classes as CIFAR-100 [...]**
>
> Experiments on CIFAR-100 are provided in Table 4.
>
> As it is suggested by the title of our paper, but also explicitly mentioned in the “Limitations” section (Lines 498-506) and MultiClass Classification section (Appendix E, Lines 774-794),  our method is designed for [and performs well when] there is limited teacher logit information, such as in binary classification tasks.  Indeed, as the number of classes increases, we anticipate LELP’s performance to converge with vanilla knowledge distillation (and we explicitly mention that in the lines mentioned above). Consequently, we did not present experiments on datasets with a very large number of classes, since LELP is not designed for such scenarios.
>
> With that being said, one should not underestimate the importance and difficulty of distillation in the binary classification (and few class classification in general), as it is one of the most common and impactful scenarios that affect major multi-billion dollar industries. Applications include:
>
> **1.Click-through prediction**: A core component of online advertising, determining ad relevance and user engagement.
> **2.Query understanding**: Essential for search engines to accurately interpret user intent and deliver meaningful results.
> **3.Sentiment analysis**: A valuable tool for gauging public opinion, informing product development, and shaping marketing strategies.
>
> Yet, as suggested both by past results (e.g., [1]) and our experiments, existing distillation techniques struggle in this setting, especially for data types beyond Computer Vision (e.g., NLP).
>
> -**I found it very hard to understand the t-SNE plots provided in Figure 4. [..]**
>
>
> #### Description
>
> Figure 4 has three columns  and two rows. Each column corresponds to t-SNE visualization of the feature embeddings learned by Vanilla KD, LELP and Oracle Clustering, respectively, during knowledge distillation from a ResNet-92 teacher to a ResNet-56 student on the binarized CIFAR-10 dataset. For a given column, the top and bottom row correspond to the same plot, colored differently:  The first row depicts embeddings colored according to the two primary classes of the task, while the second row uses color to represent the underlying subclasses (recall this is the binarized CIFAR-10 dataset, where we binary labels based on the original class labels (y_binary = y_original%2). )
>
> #### Motivation
>
> We employed t-SNE visualization as a means to qualitatively assess the feature embeddings learned by our models. The closer the proximity of two points in the t-SNE plot, the more semantically similar their interpretations are according to the model.
> As it can be observed, the 'ideal' Oracle Clustering approach yields a clear delineation of the underlying subclass structure, with distinct clusters corresponding to each subclass. This supports the hypothesis that capturing semantically meaningful subclasses contributes to improved performance.
> Furthermore, the t-SNE plot for the LELP-trained student model exhibits a richer embedding structure compared to the Vanilla KD approach. This suggests that LELP facilitates the capture of more inherent subclass information within the dataset, offering a plausible explanation for its superior performance.
>
>
> We will revise the relevant parts of the text to ensure this explanation is clearly conveyed to all readers.
>
> - **An intermediate analysis that shows the meaning of the sub-classes obtained by PCA could help for visualization and understanding. For instance, would we observe meaningful fine-grained classes?**
>
> Thank you  for your suggestion — we plan to do this in the next version of our paper. Motivated by your question, and to demonstrate that this is indeed the case, we tested to what extent the pseudo-classes discovered by each method on the binary CIFAR-10 task can predict the original classes of the CIFAR-10 dataset (we do so by directly taking the arg max of the subclass layer and picking the permutation that maximizes the accuracy in the original 10-classification problem). The results are as follows:
>
> a. Agglomerative: 32.22%
>
> b. K-means: 42.40%
>
> c. t-sne & K-means: 50.97%
>
> d. LELP (ours): 51.32%

---

> ### Author Response · Authors · 2024-11-19
> **Response to Reviewer Hun6 (part 3)**
>
> -**Main concern is the weak experimental section. Only Table 2 provides detailed classification results. Only one teacher and student architectural choices were used. It is not clear how the method generalizes to other architectures. Also, the results are not convincing enough to my opinion and in some cases are marginal.**
>
> We respectfully disagree.
>
> 1. Table 2 provides experiments with 3 types of teachers (Albert-large, Albert-XL, Albert-XXL) and two types of students  (Albert-base, MLPs operating over T5-(11B) frozen embeddings). We also include several more experiments in Tables 3 and 4 using ResNet and MobileNet architectures.  See Lines 339-356 for details.
>
> 2. Our method *consistently* performs on par, and typically outperforms the best baseline in each scenario. (No other method is so consistent).
>
> 3. Our method outperforms all other baselines by significant margins in the two largest (and most difficult) datasets considered in this study, namely  Amazon Reviews (5 classes, 500k examples) and Sentiment140 (binary, 1.6 million examples) achieving an improvement of 1.85% and 0.88%, respectively, over the best baseline. In fact, in the former case, the LELP-trained student outperforms even the teacher, which contains over 20x the number of parameters.
>
>
> - **What is the impact of the S hyper-parameter? (The number of sub-classes per class). I would expect some ablation study on this.**
>
> We have already presented ablations on the impact of the S hyper-parameter — see Figure 5 (where the horizontal axis corresponds to the number of subclasses, i.e., S), and the discussion in Appendix C (Lines 678-732).
>
> - **Minor: Line 73: form —> from. Line 202: coarse-graned —> coarse-grained**
>
> Thank you — we will make sure to address these typos in the next version of our paper.

---

> ### Author Response · Authors · 2024-11-24
> **Response to Reviewer Hun6 (part 4)**
>
> We appreciate the reviewer acknowledging our responses and increasing their score accordingly. We have responded to the reviewer's new comments below and are always happy to answer any additional concerns.
>
>
> **1a. Please note that FitNet is an extreme case of learnable projections at every layer.**
>
> Let us clarify that we implement FitNet as it is described in the original paper of Romero et. al. in [2]. Concretely, we apply Algorithm 1, Section 2.3 in [2] with h (hint layer) corresponding to the teacher's embeddings layer  and g (guided layer)  corresponding to the student's embedding layer introducing a learnable projection in the case of a dimensions-mismatch —  see also Figure 1b in [2].
>
> In other words,  we are faithful to the original description of FitNet and we are **not** introducing learnable projections at every layer.  (In our paper, we describe our implementation in  Lines 138-145, and give further details in Lines 990-994.) That said, we are aware that FitNet and related methods, including our own, can be  naturally extended by transferring knowledge from the teacher to the student using multiple teacher-layers (and that this might involve incorporating learnable projections to facilitate the transfer). However, we believe such an analysis would detract from the core message of the paper.
>
> **1b. There are other representation distillation techniques which can easily support teacher and student of different dimensions simply by adding projection layer at the final outptut of the network (as CRD for example).**
>
> We agree,  and this is exactly how we implement FitNet, CRD, VID etc in our paper. We emphasize that *our claim is not that these methods are not applicable*, rather that they do not always perform very well, especially when the teacher and student models come from very different architecture-families.
>
>
> **2a. Using FItNet may require hyper-parameter optimization (as the knowledge distillation parameter loss parameters, learning rate etc). It is unclear what effort the authors performed to make it work.**
>
> The implementation details, and the related hyperparameter optimization, of our study are presented in Appendix H.  To be fair to all methods, we either optimized hyperparameters such learning rate for standard training of the student model, i.e., without using distillation, or used Vanilla-KD-optimized hyperparameters that have been used in other papers (e.g., from Stanton et al. [3] in the case of ResNets models).
>
> While we understand that the reviewer may be skeptical about the effort we put on implementing baselines methods, we'd like to highlight that previous work, such as [1], has also shown FitNet to underperform in specific situations, including binary classification tasks, which are relevant to our research. This supports our findings and emphasizes the need for alternative approaches.
>
>
>
> **2b. Overall, from my experiment (and from other works in the literature) using different representation dimensions for teacher and student is not an issue as one can simply project them to a shared dimensions.**
>
> Again, we are not claiming that methods like FitNet etc are not applicable. We claim, and experimentally show, that they are not as effective in certain settings, as it is also suggested by [1] in the case of FitNet.
>
>  Our focus in this paper is on distillation with a limited number of classes, such as binary classification, particularly in domains beyond computer vision, like natural language processing.  We maintain that most existing distillation methods struggle in these scenarios.  However, we're open to incorporating or discussing any relevant work the reviewer might suggest.
>
>
> [1] Subclass Distillation (Rafael Müller, Simon Kornblith, Geoffrey Hinton)
>
> [2] FitNets: Hints for Thin Deep Nets. (Adriana Romero, Nicolas Ballas, Samira Ebrahimi Kahou, Antoine Chassang, Carlo GattaYoshua Bengio)
>
> [3] Does knowledge distillation really work? (Samuel Stanton, Pavel Izmailov, Polina Kirichenko, Alexander A Alemi, and Andrew G Wilson.)

---

### Official Review · Reviewer_PjQV · 2024-11-04

**Soundness:** 3
**Presentation:** 3
**Contribution:** 3
**Rating:** 5
**Confidence:** 4

**Summary:**

This manuscript proposes Learning Embedding Linear Projections, a method for distilling knowledge from a teacher model's representations. The proposed method identifies informative linear subspaces in the teacher's embedding space and converts them into pseudo-subclasses to teach students. It leverages the structure of final-layer representations and improves student performance, especially in finetuning tasks with a few classes without requiring retraining of the teacher model.

**Strengths:**

1.The idea of modality-independent of model distillation is interesting.
2.The implementation details are well-presented in the paper and comprehensive experiments are provided.
3.This paper is fluently written. The proposed method is easy to follow.

**Weaknesses:**

1.In the abstract and introduction section, the authors mentioned that existing methods can not perform well on few-class distillation because of the teacher model’s generalization patterns scales based on the number of classes. Could you explain more about this issue? Also, why the proposed method can solve the impact of poor generalization of the teacher model?
2.Most of the references on knowledge distillation are around 2020, which is too early and limited to reflect the development of work in the past two years. Also, is (Müller et al. 2020) the only research that uses sub-classes to solve few-class distillation?
3.What is the actual impact of neural collapse mentioned in the related work on few-class distillation, and what is the relationship between the proposed method and neural collapse? I am puzzled because this is not reflected in the experiments.
4.The innovation of this method is not very novel. The proposed sub-classes framework has been developed by (Müller et al. 2020), and it just adds a pseudo-subclasses splitting component by PCA decomposition compared to (Müller et al. 2020).
5.Nowadays, in large-scale scenarios, binary classification and few-class classification tasks are not commonly utilized. Are there any practical applications for studying the distillation of binary classification? For example, what is the practical significance of the binarization experiments of CIFAR-10.
6.If other related works use sub-classes for distillation, please consider citing and comparing them in the experiments.
7.As for the datasets without subclass structure (Table 2), the gain over the best baseline is minimal. Half of the experiments show an improvement of approximately 0.3% or even less.
8. The authors missed some related works that use multi-granularity class structures to address various tasks, e.g., long-tailed classification, incremental learning etc. The motivations between this manuscript and thses works are similar, so it is crucial to review such works to faciliate uncderstanding.

[1] Müller R, Kornblith S, Hinton G. Subclass distillation[J]. arXiv preprint arXiv:2002.03936, 2020.

**Questions:**

See Weaknesses.

---

> ### Author Response · Authors · 2024-11-19
> **Response to Reviewer  PjQV (part 1)**
>
> We thank the reviewer for the time and effort put in evaluating our work. We have responded to the reviewer's comments below and are happy to answer any additional questions.
>
> **1a. In the abstract  [...] Could you explain more about this issue?**
>
> In the abstract and introduction, we highlight the challenge of knowledge distillation with few classes, drawing upon the observation that the richness of the "generalization pattern" information scales with the number of classes. This concept, rooted in the findings of Müller et al., suggests that a teacher model's predictions in a high-dimensional output space (many classes) inherently convey more nuanced information about its decision-making process compared to a low-dimensional output space (few classes).
>
> To illustrate, consider the difference between a 1000-class and a 2-class classification problem. The teacher's output in the 1000-class case (e.g., a probability distribution over 1000 classes) carries significantly more information about its learned features and generalization strategies than in the 2-class case. This richer information allows for more effective knowledge transfer to the student.
>
> Existing distillation methods often struggle with few-class scenarios because they are less effective at capturing and transferring this limited "generalization pattern" information. Our proposed method addresses this issue by inventing informative pseudo-subclasses that enrich the output space and enhance the transfer of the teacher's generalization patterns to the student.
>
> In essence, our method aims to extract and distill the crucial knowledge embedded within the teacher's predictions, even when the number of classes is limited, thus enabling effective learning in few-class knowledge distillation.
>
> We will revise the relevant sections to ensure this explanation is clearly conveyed to all readers.
>
> **1b. Also, why the proposed method can solve the impact of poor generalization of the teacher model?**
>
> We are not making such a general claim. Perhaps the reviewer is referring to our experiments on the semi-supervised distillation setting (see Appendix F for more details). We suspect that our  stronger performance in this setting may stem from the inherent label smoothing effects of our distillation loss.
>
>
> **2. Most of the references on knowledge distillation are around 2020 [..]?**
>
> We have made every effort to include and compare against all relevant distillation methods applicable to our specific setting. However, we are always open to suggestions and would be happy to incorporate any additional methods that the reviewer deems relevant for a more comprehensive analysis.
>
> **3. What is the actual impact of neural collapse [... ] ?**
>
> As we describe in Lines 197-209, our method is inspired by the recent paper of Yang et al. on the aspects of the neural collapse phenomenon. In a nutshell:
>
> - Yang et al. show both theoretically and experimentally that while the final layer representations (embeddings) appear collapsed, they retain crucial fine-grained structure. They show that by applying unsupervised clustering techniques to the model’s internal representations, they are able to discover semantically meaningful pseudo-subclasses.
> - Creating meaningful pseudo-subclasses aids student knowledge transfer as revealed both by our experiments (Section 4.2) and the work of Müller et al.
> - We present an unsupervised technique to extract knowledge from teacher embeddings using linear projections to form pseudo-subclasses.
>
> **4. The innovation of this method is not very novel [..].**
>
> While we certainly agree that our work is heavily inspired by Müller et al., we would like to emphasize the two following points:
>
> - As we mention in Lines 151-161, the Subclass Distillation method of Müller et al. is essentially not applicable in real-world scenarios, as it requires retraining the teacher model several times (once for each hyperparameters-configuration).  Moreover, in many real-world settings, one is only given inference-access to the teacher model, i.e., one is not allowed to retrain it (let alone several times!)
>
> - Our method often significantly outperforms Subclass Distillation (and all other methods) in large-scale settings such as Amazon Reviews (5 classes, 500k examples) and Sentiment140 (binary, 1.6 million examples) achieving an improvement of 1.85% and 0.88%, respectively, over the best baseline.

---

> ### Author Response · Authors · 2024-11-19
> **Response to Reviewer PjQV (part 2)**
>
> **5a. [...] binary classification and few-class classification tasks are not commonly utilized [...]**
>
> On the contrary, binary classification is arguably one of the most widespread and impactful classification types used today. Its applications are far-reaching and crucial to the success of major multi-billion dollar industries. For example:
>
> - **Click-through prediction**: A core component of online advertising, determining ad relevance and user engagement.
> - **Query understanding**: Essential for search engines to accurately interpret user intent and deliver meaningful results.
> - **Sentiment analysis**: A valuable tool for gauging public opinion, informing product development, and shaping marketing strategies.
>
> Yet, as suggested both by past results (e.g., [1]) and our experiments, existing distillation techniques struggle in this setting, especially for data types beyond Computer Vision (e.g., NLP).
>
> **5b. For example, what is the practical significance of the binarization experiments of CIFAR-10.**
>
>
> The experiments on binarized CIFAR-10/100 are not presented for their practical significance (we have experiments on Amazon Reviews and Sentiment140 for this purpose).
>
> The main reason we considered the binarized CIFAR-10/100 experiments is that the inherent subclass structure of these settings makes them ideal for exploring several natural ways of inventing pseudo-subclasses via clustering and showing that this approach can enhance distillation performance. Crucially, as we mention in Line 407, the availability of original labels in these  datasets allows us to explore the "Oracle Clustering" approach, where the subclass structure is known a priori. This serves as an upper bound, and emphatically suggests that the generation of pseudo-subclasses through clustering techniques has substantial promise.
>
> **6. If other related works use sub-classes for distillation [...]**
>
> We have made every effort to include and compare against all relevant distillation methods applicable to our specific setting. Besides the original Subclass distillation paper, we have not found any other paper that is applicable to the scenarios considered in the paper (and is substantially different from Subclass distillation).
>
> That said, we will include to our reference list papers that make use variations of subclass distillation for specialized settings.
>
> Also, and as we have already said, we are always open to suggestions and would be happy to incorporate any additional methods that the reviewer deems relevant for a more comprehensive analysis.
>
>
> **7. As for the datasets without subclass structure (Table 2) [...]**
>
> We would like to emphasize that:
>
> - Our method consistently performs on par with, and typically outperforms, the best baseline in each scenario. (No other method is so consistent).
>
> - Our method outperforms all other baselines by significant margins in the two largest (and most difficult) datasets considered in this study, namely Amazon Reviews (5 classes, 500k examples) and Sentiment140 (binary, 1.6 million examples) achieving an improvement of 1.85% and 0.88%, respectively, over the best baseline.
>
> **8. The authors missed some related works that use multi-granularity class [...]**
>
> Thank you for highlighting the relevance of multi-granularity class structures — we will make sure to refer to this line of work in the next version of our paper. While our initial exploration didn't reveal any applications within the specific knowledge distillation context we're addressing, we're certainly open to expanding our perspective.  We would be very grateful if the reviewer could point us towards any relevant works in this area. We are committed to thoroughly investigating this further and incorporating any pertinent references into our paper.

---

### Meta-Review · Area_Chair_5o1E · 2024-12-20

**Metareview:**

This paper aims to handle binary or few-class classification when applying knowledge distillation. To achieve this goal, the authors increase the number of classes by projecting the final embedding vectors of the student and the teacher into several PCA subsets. In this way, the teacher could contain more information that is distributed over many more effective classes, and thus transfer to the student during the training process.

Pros:

- This paper is well-written and easy to follow.
- The motivation is clear. The authors enlarge the number of classes by mapping the teacher's embedding into different subspaces to generate pseudo labels.


Reasons to reject:

- The novelty of the proposed method is limited. The proposed sub-classes framework has been developed by (Müller et al. 2020), and it just adds a pseudo-subclasses splitting component by PCA decomposition compared to (Müller et al. 2020).
- The experimental performance is not promising. We can see that the improvements over compared methods are marginal.
- The generalizability of the proposed method is low. Binary classification and few-class classification tasks are not commonly utilized in many practical scenarios. We expect that it can handle more real-world applications with abundant number of classes, instead of only binary / few-class classification.

**Additional Comments On Reviewer Discussion:**

This paper finally receives the scores of 6, 6, 5, 5. I have carefully checked all the comments from the reviewers and the rebuttal from the authors. Although the authors indeed have addressed some of the reviewers' concerns, I agree with Reviewer PjQV that the novelty of the proposed method and the generalizability of the proposed method are limited. I also agree with Reviewer Hun6 that the practical improvements are marginal.

Considering the above situations and given that the overall score of this paper is relatively low, I have to recommend rejecting this paper.

---

### Decision · Program_Chairs · 2025-01-22

Reject